# EFFICIENT QUANTIZATION-AWARE ADAPTATION FOR VISUAL FOUNDATION MODELS

## ABSTRACT

Efficient strategies to jointly adapt and deploy large language models have seen a growing need under resource-limited conditions for downstream applications. However, when applied to visual foundation models, existing methods typically incur either high GPU memory consumption during adaptation or extra computation costs introduced by the adapters at deployment. In this paper, we propose **E**fficient **Qu**antization-aware **A**daptation (EQuA) that achieves high efficiency in both adaptation and deployment for visual foundation models. We observe that dominant memory consumption arises from intermediate activations cached for backpropagation in the deep backbone and activation quantizers. To address this issue, we split a lightweight sub-network from the backbone during adaptation as a side adapter branch, and tailor two adaptation strategies to eliminate these cached activations, thereby significantly reducing memory consumption. At deployment, the side adapter branch is merged back into the backbone, yielding a quantized model without any extra computation costs. Extensive experiments on representative visual foundation models and diverse downstream tasks exhibit that EQuA achieves an elegant trade-off between performance and efficiency. For example, EQuA yields over 70% GPU memory reduction compared to state-of-the-art baselines while maintaining competitive performance.

## 1 INTRODUCTION

Foundation models have recently exhibited impressive performance across diverse tasks. Pre-trained on massive datasets, these models can be effectively adapted, *i.e.*, fine-tuned, for diverse downstream applications. Thanks to the strong generalization ability of foundation models, there is a growing need for an efficient solution that enables their adaptation and quantization under resource-limited conditions for downstream deployment.

In the field of Large Language Models (LLMs) (Touvron et al., 2023a), some recent methods (Zhang et al., 2024; Kim et al., 2023) leverage parameter-efficient fine-tuning (PEFT) techniques to enable joint quantization and adaptation with minimal trainable parameters, avoiding the overhead of full-parameter fine-tuning. These methods typically quantize massive pre-trained floating-point weights (*e.g.*, 138.0 GB for floating-point LLaMA2-70B (Touvron et al., 2023b)) into low-bit integers (*e.g.*, 35.3 GB for 4-bit LLaMA2-70B) to reduce training memory consumption on GPU and then apply adaptation strategies via adapters.

While being effective for LLMs, however, these methods exhibit limited efficiency on visual foundation models. LLMs process token sequences and substantial memory consumption comes from pre-trained floating-point weights. In contrast, visual foundation models typically consume far less GPU memory for floating-point weights compared to LLMs (*e.g.*, 2.4 GB for SAM-H (Kirillov et al., 2023) *vs.* 138.0 GB for LLaMA2-70B), but they often process high-resolution images and perform dense prediction tasks, which generate large intermediate activations. As a result, cached activations during backpropagation become the main source of memory costs, further exacerbated by deep backbones and activation quantizers in the backward pass. As shown in Fig. 1a, QA-LoRA (Xu et al., 2024) fine-tunes only about 1.0% of the parameters in SAM-H (Kirillov et al., 2023), but the GPU memory costs during adaptation exceeds 60 GB, making it infeasible to perform joint quantization and adaptation of SAM-H on a single RTX 3090 GPU (24 GB). Although QST (Zhang et al., 2024) reduces memory consumption by blocking gradient flow through the quantized backbone via

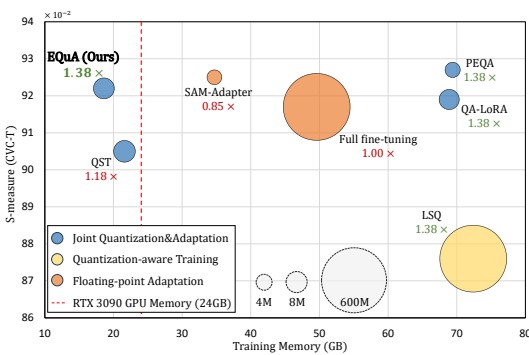 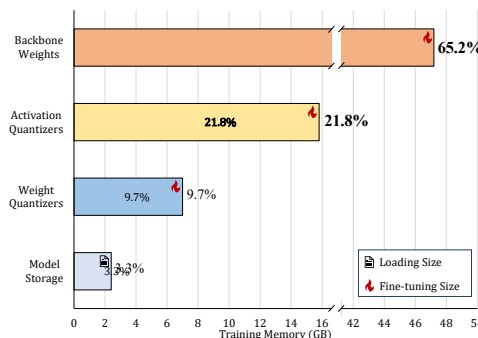

(a) Performance-efficiency trade-off  (b) Training memory components on SAM-H

Figure 1: (a) Performance-efficiency trade-off for quantizing and adapting SAM-H (2.4 GB) on the CVC-T dataset (Vázquez et al., 2016) under the 4-bit setting. The bubble size represents number of trainable parameters. The number below the method name denotes inference speedup compared to the original floating-point model. (b) The analysis is conducted on SAM-H with a batch size of 2. *Fine-tuning Size* represents memory consumption caused by fine-tuning weight quantizers, activation quantizers, or backbone weights. *Loading Size* represents the memory consumption occupied by loading the model on GPU. Total memory consumption is 72.4 GB.

side adapters, these adapters introduce extra computations and inherit quantization errors from the quantized backbone, limiting both the inference efficiency and performance of the quantized model.

In this work, we aim to design an efficient joint quantization and adaptation method for visual foundation models that reduces training memory consumption during adaptation and incurs no extra computational overhead at deployment. To this end, we propose a novel method, **E**fficient **Qu**antization-aware **A**daptation (EQuA). We first evaluate weight importance by scaling the magnitude of each weight with the corresponding input activation and split important weights from the backbone to construct a lightweight Sub-network Side Adapter (SSA) branch, while the less important weights remain in the Memory-Intensive Backbone (MIB) branch. During adaptation, we propose a Side Adapter Quantization-aware Fine-tuning (SAQF) strategy by freezing MIB and fine-tuning SSA end to end, eliminating intermediate activations cached in MIB and substantially reducing memory usage. We also propose a Block-wise Activation Quantization Fine-tuning (BAQF) strategy to optimize activation quantizers by minimizing block-wise quantization errors. SAQF and BAQF are executed alternately, alleviating the memory consumption and quantization errors exacerbated by the deep backbone and activation quantizers. At deployment, we merge the SSA with the MIB, yielding a fully quantized model with no extra computational overhead.

We evaluate the effectiveness of our EQuA on representative visual foundation models including SAM (Kirillov et al., 2023) and ViT (Dosovitskiy et al., 2021). We conduct experiments on SAM across a diverse set of downstream tasks, including medical images, natural images, and agricultural images. For ViT, we evaluate on the VTAB (Zhai et al., 2019) benchmark, which comprises 19 datasets spanning three categories: Natural, Specialized, and Structured. Extensive experiments demonstrate that our EQuA achieves an elegant trade-off between performance and efficiency. For example, the 4-bit SAM model adapted and quantized with EQuA reduces memory consumption by over 70% during adaptation compared to state-of-the-art joint quantization and adaptation methods, while maintaining competitive performance. Furthermore, the quantized SAM model achieves up to $1.38\times$ inference speedup compared to the original floating-point model.

The main contributions of this work are as follows:

1) We propose a novel method, EQuA, to achieve efficient joint quantization and adaptation. To the best of our knowledge, this is the first work specifically designed for visual foundation models.

2) We design a unique side adapter, SSA, by splitting a sub-network from the backbone, which can save GPU memory during adaptation and can be merged back into the backbone at deployment.

3) We tailor two adaptation strategies, SAQF and BAQF, to enable efficient joint quantization and adaptation in a quantization-aware manner, while preserving memory efficiency and mitigating quantization errors.

4) Extensive experiments demonstrate that our EQuA achieves a superior performance-efficiency trade-off for visual foundation models compared with existing methods.

## 2 RELATED WORK

**Adaptation for Foundation Models.** Foundation models, such as GPT-3 (Brown et al., 2020) and BERT (Devlin et al., 2019), have transformed natural language processing by enabling strong generalization across diverse tasks through large-scale pre-training. This paradigm has extended to the visual domain, resulting in powerful visual foundation models based on Vision Transformers (ViTs), including ViT-Base (Dosovitskiy et al., 2021), CLIP (Radford et al., 2021), Segment Anything Model (SAM) (Kirillov et al., 2023), and Stable Diffusion (Rombach et al., 2022). Pre-trained on large-scale image datasets, these visual foundation models exhibit strong performance across classification, segmentation, detection, and generation tasks. To adapt them efficiently, parameter-efficient fine-tuning (PEFT) techniques, such as Adapter (Chen et al., 2023; Zhang et al., 2024; Houlsby et al., 2019), LoRA (Hu et al., 2022; Zhong et al., 2024), and Prompt Tuning (Jia et al., 2022), have been proposed, which insert or adjust small trainable components while keeping the backbone frozen. However, deploying such models on edge devices requires jointly addressing adaptation and quantization challenges. In this work, we propose an efficient quantization-aware adaptation method tailored for visual foundation models.

**Quantization.** Quantization is a widely used compression and acceleration technique for deploying models on resource-constrained devices such as smartphones and TVs, by converting floating-point weights and activations into low-bit integers. It typically includes quantization-aware training (QAT) and post-training quantization (PTQ). QAT (Esser et al., 2020; Shin et al., 2023) optimizes both model weights and quantizers to achieve near full-precision accuracy, but is computationally expensive for large-scale foundation models. PTQ (Li et al., 2023; Xiao et al., 2023; Lin et al., 2024) is more lightweight, requiring only a small calibration set, but lacks end-to-end adaptation for downstream tasks and suffers from larger errors at low bit widths. In this work, we aim to enable efficient joint quantization and adaptation process for visual foundation models.

**Joint Quantization and Adaptation.** Joint quantization and adaptation (Chen et al., 2024; Dettmers et al., 2023), which integrates quantization into PEFT techniques, has attracted increasing attention. QLoRA (Dettmers et al., 2023) compresses foundation models to 4-bit precision and recovers performance with LoRA (Hu et al., 2022). QA-LoRA (Xu et al., 2024) merges LoRA parameters into quantization parameters for efficient deployment. PEQA (Kim et al., 2023) directly fine-tunes quantization parameters on downstream tasks. These methods are significant for LLMs where memory usage is dominated by floating-point weights. In visual foundation models, however, memory is dominated by activations cached for backpropagation in quantized deep backbones and activation quantizers, which limits the significance of existing methods. Although QST (Zhang et al., 2024) mitigates this issue by attaching side adapters to a quantized backbone, reducing memory from backward gradients, these adapters introduce extra computation and inherit quantization errors from the backbone, limiting inference efficiency and performance. In this work, our method introduces no extra computation and achieves both memory-efficient adaptation and inference-efficient deployment for visual foundation models.

## 3 MOTIVATION

Visual foundation models with activation quantization face significant training memory challenges during adaptation. As shown in Fig. 1b, we conduct a preliminary experiment on SAM-H, a representative visual foundation model with numerous parameters, by fine-tuning its weights and quantizers under a quantization setting. We observe that floating-point weights of SAM-H occupy only 3.3% of total GPU memory, making mainstream strategies (*e.g.*, QA-LoRA (Xu et al., 2024)) in LLMs that save memory by quantizing the backbone largely ineffective for visual foundation models. Instead, the dominant consumption arises from two sources. First, fine-tuning the backbone weights alone consumes over 45 GB because multiple memory-intensive ViT blocks of the deep backbone cache large amounts of intermediate activations for backpropagation. Such observation is consistent with prior studies (Mercea et al., 2024). Second, fine-tuning numerous activation quantizers introduces additional cached activations, further leading to an extra 21.8% memory consumption.

QST (Zhang et al., 2024), an inspiring solution, reducing memory by attaching additional side adapters to a quantized LLM backbone. However, these side adapters introduce extra computation costs, limiting inference efficiency, as shown in Fig. 1a. Moreover, QST doesn't optimize quantizers. In vision tasks, the activation quantization is often sensitive and incurs substantial quantization errors. These errors accumulate throughout the backbone and are absorbed by the side adapters, degrading the performance of visual foundation models. Further analysis is provided in Appendix D.

In the following section, we design EQuA, which addresses memory issues and preserves performance in a quantization-aware manner while incurring no additional computation during inference.

## 4 METHOD

### 4.1 PRELIMINARIES

**Uniform Quantization.** Uniform quantization is commonly used in various quantization methods due to its compatibility with hardware acceleration. The formulation of $b$-bit uniform quantization of the floating-pint value $x$ is as follows:

$$Quant(x, s, z): \quad x_q = clip(\lfloor \frac{x}{s} \rceil + z, 0, 2^b - 1), \tag{1}$$

$$Dequant(x_q, s, z): \quad \hat{x} = s \cdot (x_q - z) \approx x,$$

where $x_q$ and $\hat{x}$ denote the quantized and de-quantized values, respectively. During inference, $\hat{x}$ can be replaced by $x_q$ to enable integer-only computation (Jacob et al., 2018). $\lfloor \cdot \rceil$ denotes the rounding function, and $clip(x, l, u) = min(max(x, l), u)$. The scaling factor $s$ and zero point $z$ are computed from the observed lower and upper bounds of $x$:

$$s = \frac{x_{ub} - x_{lb}}{2^b - 1}, \quad z = \lfloor -\frac{x_{ub}}{s} \rceil. \tag{2}$$

Using separate $s$ and $z$ for each channel defines channel-wise quantization, while sharing single $s$ and $z$ across the all channels defines tensor-wise quantization. In this work, We apply the channel-wise quantization on weights and the tensor-wise quantization on activations.

**ViT Block.** Visual foundation models are mainly based on the ViT block (Dosovitskiy et al., 2021), which consists of a Multi-Head Self-Attention (MHSA) module and a Multi-Layer Perceptron (MLP) module. The formulas are below:

$$Y_{l-1} = MHSA(LN(F_{l-1})) + F_{l-1}, \tag{3}$$

$$F_l = MLP(LN(Y_{l-1})) + Y_{l-1},$$

where $LN$ is layer normalization. $F_{l-1}/F_l$ denote input/output of $l$-th ViT block. Here, the bias term is omitted for simplicity. The calculation of MHSA module is as follows:

$$MHSA(X) = AXW^V W^{Proj}, \tag{4}$$

where $A$ denotes the attention map and is calculated by $W^Q$ and $W^K$. $W^Q, W^K, W^V \in \mathbb{R}^{D \times HD_h}$ and $W^{Proj} \in \mathbb{R}^{D \times D}$. $H$ is the number of attention heads and $D_h$ is the dimension size of each head, where $D = D_h \cdot H$. The calculation of MLP module is as follows:

$$MLP(X) = GeLU(XW^{Lin1})W^{Lin2}, \tag{5}$$

where $W^{Lin1} \in \mathbb{R}^{D \times D_r}$ and $W^{Lin2} \in \mathbb{R}^{D_r \times D}$. $D_r$ is the hidden dimension size.

### 4.2 SUB-NETWORK SIDE ADAPTER

To mitigate the substantial memory overhead from the deep backbone of visual foundation models during adaptation and avoid additional computations and parameters during deployment, we design an SSA inspired by recent works (Zhang et al., 2024; Jiang et al., 2023). We split a lightweight sub-network from each of the two modules in Eq. 3 to construct the SSA for the ViT block. The MHSA module can be split into two terms, which correspond to the MIB and SSA branches, respectively:

$$MHSA(X) = AX \begin{bmatrix} W_{mib}^V & W_{ssa}^V \end{bmatrix} \begin{bmatrix} W_{mib}^{Proj} \\ W_{ssa}^{Proj} \end{bmatrix} = \underbrace{AXW_{mib}^V W_{mib}^{Proj}}_{MHSA_{mib}(X)} + \underbrace{AXW_{ssa}^V W_{ssa}^{Proj}}_{MHSA_{ssa}(X)}, \tag{6}$$

where $W_{mib}^V \in \mathbb{R}^{D \times (D-D_s)}$ and $W_{mib}^{Proj} \in \mathbb{R}^{(D-D_s) \times D}$ represent weights of linear layer in the backbone branch, and $W_{ssa}^V \in \mathbb{R}^{D \times D_s}$ and $W_{ssa}^{Proj} \in \mathbb{R}^{D_s \times D}$ represent weights of linear layer in the SSA branch. The attention map $A$ is shared in $MHSA_{mib}$ and $MHSA_{ssa}$. $W^Q$ and $W^K$

Figure 2: Overview of our EQuA. During quantization-aware adaptation for downstream tasks, we split each ViT block into a Memory-Intensive Backbone (MIB) branch and a Sub-network Side Adapter (SSA) branch. To reduce memory consumption, the computation graph is constructed only for the SSA branch to cache intermediate activations. At deployment, the two branches are merged back into the original ViT block, introducing no additional computational overhead. $\hat{W}$ represents weights of linear layer are quantized. The split layers share the same activation quantizer.

cannot be expressed in the block-matrix form described in Eq. 6, and are therefore kept unsplit. During backpropagation, we update $W_{ssa}^V$ and $W_{ssa}^{Proj}$, while preventing gradient flow through the path involving $MHSA_{mib}$ and attention map $A$. The MLP module can also be split into two terms:

$$MLP(X) = GeLU(X \begin{bmatrix} W_{mib}^{Lin1} & W_{ssa}^{Lin1} \end{bmatrix}) \begin{bmatrix} W_{mib}^{Lin2} \\ W_{ssa}^{Lin2} \end{bmatrix}$$

$$= \underbrace{GeLU(XW_{mib}^{Lin1})W_{mib}^{Lin2}}_{MLP_{mib}(X)} + \underbrace{GeLU(XW_{ssa}^{Lin1})W_{ssa}^{Lin2}}_{MLP_{ssa}(X)}, \qquad (7)$$

where $W_{mib}^{Lin1} \in \mathbb{R}^{D \times (D_r - D_s)}$, $W_{mib}^{Lin2} \in \mathbb{R}^{(D_r - D_s) \times D}$, $W_{ssa}^{Lin1} \in \mathbb{R}^{D \times D_s}$, and $W_{ssa}^{Lin2} \in \mathbb{R}^{D_s \times D}$. $D_s$ is the SSA dimension. We update $W_{ssa}^{Lin1}$ and $W_{ssa}^{Lin2}$ and freeze $MLP_{mib}$.

We propose a strategy to split the SSA from the original linear layer based on weight importance, assuming that weight channels with higher importance have greater impact on performance. A simple and effective way to estimate weight importance is by measuring the magnitude of the weights, as channels with larger magnitudes are generally more important (Han et al., 2015). However, for visual foundation models adapted to downstream tasks, input activation distributions may shift across domains. Inspired by existing work (Sun et al., 2024), we scale weight magnitudes by input activations to better capture the relative importance of each channel for downstream adaptation. Consider an input activation $X$ of shape $(B \times N, D_{in})$ and a linear layer with weight $W \in \mathbb{R}^{D_{in} \times D_{out}}$, where $B$ and $N$ are batch size and sequence length, respectively. The scaled magnitude used to estimate weight importance is defined as follow. Detailed mathematical derivation about weight importance is provided in Appendix A.

$$M_i = \sum_{j=1}^{D_{out}} \|X_{:,i}\|_2 \cdot |W_{i,j}|, \quad P_i^M = Softmax(M_i), \qquad (8)$$

where $M_i$ denotes the weight importance of $i$-th input channel, $P_i^M$ denotes the probability of importance, $|\cdot|$ represents the absolute value function, and $\|X_{:,i}\|_2$ denotes the $L_2$ norm computed over the $i$-th channel across all $B \times N$ tokens. We apply Eq. 8 to $W^{Proj}$ and $W^{Lin2}$ to compute their respective weight importance probabilities, guided by which a subset of channels, referred to as SSA channels, is stochastically selected to construct the SSA:

$$MHSA_{ssa}: \quad W_{ssa}^V = W_{:,\mathcal{I}_{ssa}}^V, \qquad W_{ssa}^{Proj} = W_{\mathcal{I}_{ssa},:}^{Proj};$$

$$MLP_{ssa}: \quad W_{ssa}^{Lin1} = W_{:,\mathcal{I}_{ssa}}^{Lin1}, \qquad W_{ssa}^{Lin2} = W_{\mathcal{I}_{ssa},:}^{Lin2}, \qquad (9)$$

where $\mathcal{I}_{ssa}$ denotes a set of indices of SSA channels. In practice, $\mathcal{I}_{ssa}$ is sampled according to the corresponding probabilities $P_i^M$, rather than by selecting the top-$D_s$ $M_i$, for a more stable convergence. Since the original linear layer weights are split into $W_{ssa}$ and $W_{mib}$, they can be merged at deployment to obtain a fine-tuned weight with the same shape as the original one, without introducing extra parameters. Further analysis about SSA channel selection is provided in Appendix B.

### 4.3 SIDE ADAPTER QUANTIZATION-AWARE FINE-TUNING STRATEGY

Fig. 2 illustrates the SAQF process of our EQuA. During the adaptation phase, each ViT block is split into a backbone branch and an SSA branch. We fine-tune weights and weight quantizers of the quantized linear layers in the SSA branch while keeping the MIB branch frozen. Since the computation graph is constructed only for the low-dimensional SSA branch during adaptation, which caches low-dimensional intermediate activations, and the high-dimensional MIB branch is excluded from the graph construction, this results in substantial memory savings. To further facilitate convergence, we apply gradient checkpointing to the input activations of the $W_{mib}^{V}$ and $W_{mib}^{Lin1}$ layers in the backbone branch. Their gradients can be recomputed on the fly during backpropagation without incurring significant memory overhead and merged into the gradient flow of SSA branch, enabling SSA to learn quantization-aware information about quantized MIB and thereby improving performance, as verified in Table 3. At deployment, the SSA and MIB branches are merged into a single ViT block, with all linear layers preserving the shape of the original model. This design introduces no additional parameters or computations, thereby maintaining inference efficiency. More details about gradient checkpointing can be found in Appendix E.

To further reduce trainable parameters, we introduce LoRA (Hu et al., 2022) into the linear layers of the SSA branch, enabling a low-rank quantization-aware adaptation:

$$\hat{W}_{ssa} = Dequant(Quant(W_{ssa} + \alpha \cdot A_r B_r, s_{w_{ssa}}, z_{w_{ssa}}), s_{w_{ssa}}, z_{w_{ssa}}), \tag{10}$$

where $Dequant(Quant(\cdot))$ is the uniform quantization in Eq. 1, $s_{w_{ssa}}$ and $z_{w_{ssa}}$ are scaling factors and zero points of $W_{ssa}$, respectively, $A_r \in \mathbb{R}^{D \times r}$ and $B_r \in \mathbb{R}^{r \times D_s}$ are low-rank matrices, and $\alpha$ is a scalar. During adaptation, we freeze $W_{ssa}$ and fine-tune $A_r$, $B_r$, and $s_{w_{ssa}}$.

### 4.4 BLOCK-WISE ACTIVATION QUANTIZATION FINE-TUNING STRATEGY

We design the BAQF strategy to mitigate quantization errors and reduce memory consumption incurred by activation quantizers in the backbone. After merging the SSA and MIB branches into a unified ViT block, we perform BAQF by minimizing the Mean Squared Error (MSE) between the outputs of each quantized block $\mathcal{F}_l$ and the corresponding block $\mathcal{F}_l^{fp}$ with quantizers disabled:

$$s_a \leftarrow s_a - \eta \cdot \nabla_{s_a} \mathbb{E}[\|\mathcal{F}_l^{fp}(Y_{l-1}) - \mathcal{F}_l(\hat{Y}_{l-1})\|_2] \tag{11}$$

where $Y_{l-1}$ and $\hat{Y}_{l-1}$ denote floating-point input and de-quantied input of $l$-th block, respectively, $s_a$ denotes scaling factors of activation quantizers. Here, we update only $s_a$ while keeping all other parameters frozen. Since the BAQF strategy operates in a block-wise manner, the memory consumption introduced by activation quantizers is negligible, as verified in Table 3. We perform BAQF once every $K$ epochs of SAQF, where $K$ denotes the block-wise fine-tuning frequency. More details on the EQuA processing pipeline are provided in Appendix G.

## 5 EXPERIMENTS AND RESULTS

### 5.1 EXPERIMENTAL SETUP

**Settings.** For the evaluation of our EQuA, we perform downstream quantization and adaptation experiments on two representative visual foundation models, SAM-Huge (SAM-H) (Kirillov et al., 2023) and ViT-Base (ViT-B) (Dosovitskiy et al., 2021), both of which are pre-trained on large scale datasets. We first use RepQ-ViT (Li et al., 2023) to perform uniform quantization on the pre-trained models as a PTQ initialization by jointly calibrating quantization parameters on the original full-precision weights, and then fine-tune the quantized models on downstream datasets. We include different bit-width configurations supported by the hardware, including W8A8 (8-bit weights and activations) and W4A4 (4-bit weights and activations). We fine-tune quantized pre-trained models using the Adam (Kingma & Ba, 2015) and the CosineAnnealingLR (Loshchilov & Hutter, 2017).We set the batch size to 2 for SAM-H and 32 for ViT-B. The SSA dimension $D_s$ (Eq. 7) is set to 64, the LoRA rank $r$ (Eq. 10) is set to 4, the scalar $\alpha$ (Eq. 10) is set to 32 for SAM-H and 8 for ViT-B, and $K$ (Sec. 4.4) is set to 10 for SAM-H and 20 for ViT-B. More detailed setups are in Appendix F.

**Datasets and Metrics.** Following recent studies (Zhong et al., 2024; Chen et al., 2023), we conduct experiments on SAM-H across three downstream semantic segmentation scenarios: polyp segmentation for medical images (Vázquez et al., 2016; Jha et al., 2020; Silva et al., 2014), camouflaged

Table 1: Semantic segmentation results of SAM-H on various downstream datasets. *FP* denotes floating-point methods. *Param* and *Mem* indicate the number of trainable parameters and training memory consumption, respectively. The best, second-best, and third-best results are marked in **bold**, underlined, and double underlined, respectively. The ⌀ denotes $Param < 10M$, and 🍂 indicates that the model can be trained on a single RTX 3090 GPU, *i.e.*, $Mem < 24GB$.

| Method | Bit | Param↓ | Mem↓ | Medical | | | | | | Natural | | | | Agricultural | |
|---|---|---|---|---|---|---|---|---|---|---|---|---|---|---|---|
| | | | | CVC-T | | Kvasir | | ETIS | | CAMO | | COD10K | | Leaf | |
| | | | | $S_\alpha$↑ | $E_\phi$↑ | $S_\alpha$↑ | $E_\phi$↑ | $S_\alpha$↑ | $E_\phi$↑ | $S_\alpha$↑ | $E_\phi$↑ | $S_\alpha$↑ | $E_\phi$↑ | mIoU↑ | mDice↑ |
| Full | FP | 641.09 M | 49.6 GB | 0.917 | 0.940 | 0.917 | 0.954 | 0.815 | 0.833 | 0.805 | 0.860 | 0.837 | 0.892 | 0.710 | 0.818 |
| SAM-Adapter | FP | 4.25 M | 34.7 GB | 0.925 | 0.939 | 0.915 | 0.941 | 0.829 | 0.838 | 0.867 | 0.903 | 0.894 | 0.930 | 0.671 | 0.780 |
| LoRA | FP | 8.03 M | 45.4 GB | 0.937 | 0.963 | 0.936 | 0.962 | 0.845 | 0.857 | 0.889 | 0.933 | 0.920 | 0.959 | 0.730 | 0.831 |
| LSQ | W8A8 | 641.46 M | 72.4 GB | 0.920 | 0.958 | 0.900 | 0.937 | 0.793 | 0.825 | 0.808 | 0.864 | 0.843 | 0.900 | 0.712 | 0.818 |
| NIPQ | W8A8 | 641.46 M | 72.4 GB | 0.916 | 0.956 | 0.904 | 0.936 | 0.797 | 0.819 | 0.805 | 0.859 | 0.841 | 0.898 | 0.702 | 0.811 |
| QA-LoRA | W8A8 | ⌀ 7.13 M | 68.9 GB | 0.921 | 0.936 | **0.930** | **0.953** | **0.865** | **0.895** | **0.885** | **0.928** | **0.919** | **0.959** | 0.720 | 0.823 |
| PEQA | W8A8 | ⌀ 4.43 M | 69.4 GB | **0.929** | **0.969** | 0.929 | 0.953 | 0.827 | 0.844 | 0.877 | 0.923 | 0.909 | 0.951 | **0.732** | **0.835** |
| QST | W8A8 | ⌀ 8.30 M | 🍂 21.6 GB | 0.918 | 0.939 | 0.899 | 0.925 | 0.799 | 0.819 | 0.839 | 0.884 | 0.885 | 0.930 | 0.678 | 0.792 |
| EQuA (Ours) | W8A8 | ⌀ 7.67 M | 🍂 18.5 GB | 0.925 | 0.961 | 0.919 | 0.948 | 0.834 | 0.848 | 0.863 | 0.900 | 0.902 | 0.942 | 0.717 | 0.820 |
| LSQ | W4A4 | 641.46 M | 72.4 GB | 0.876 | 0.895 | 0.895 | 0.931 | 0.753 | 0.769 | 0.760 | 0.804 | 0.806 | 0.867 | 0.697 | 0.807 |
| NIPQ | W4A4 | 641.46 M | 72.4 GB | 0.884 | 0.908 | 0.901 | 0.934 | 0.771 | 0.791 | 0.762 | 0.808 | 0.818 | 0.875 | 0.699 | 0.809 |
| QA-LoRA | W4A4 | ⌀ 7.13 M | 68.9 GB | 0.919 | 0.929 | 0.923 | 0.948 | 0.830 | 0.862 | 0.858 | 0.903 | 0.903 | 0.945 | 0.705 | 0.815 |
| PEQA | W4A4 | ⌀ 4.43 M | 69.4 GB | 0.927 | 0.959 | 0.926 | 0.952 | 0.823 | 0.842 | 0.855 | 0.899 | 0.892 | 0.937 | 0.718 | 0.819 |
| QST | W4A4 | ⌀ 8.30 M | 🍂 21.6 GB | 0.905 | 0.915 | 0.894 | 0.921 | 0.729 | 0.754 | 0.788 | 0.836 | 0.833 | 0.884 | 0.656 | 0.772 |
| EQuA (Ours) | W4A4 | ⌀ 7.67 M | 🍂 18.5 GB | 0.922 | 0.938 | 0.915 | 0.947 | 0.799 | 0.837 | 0.833 | 0.878 | 0.873 | 0.920 | 0.709 | 0.816 |

|  |  |  |  |  |  |
|---|---|---|---|---|---|
| Input | QA-LoRA | PEQA | QST | EQuA (Ours) | GT |

Figure 3: Visualization results of SAM-H on semantic segmentation under the W4A4 setting, with cases from three different downstream datasets: Kvasir, CAMO, and Leaf.

object detection for natural images (Le et al., 2019; Fan et al., 2020a), and leaf disease segmentation for agricultural images (Zhong et al., 2024). For evaluation of semantic segmentation results, we use the widely-used S-measure ($S_\alpha$) and mean E-measure ($E_\phi$) for medical images and natural images, and use mean IoU (mIoU) and mean Dice (mDice) for agricultural images. We also conduct experiments on ViT-B using the VTAB (Zhai et al., 2019) benchmark, which comprises 19 image classification datasets across three categories and is widely adopted in the vision community (Mercea et al., 2024; Jiang et al., 2023). We use the Top-1 accuracy to evaluate image classification results. More details about datasets and metrics are provided in Appendix H.

**Baselines.** We include following baselines: 1) floating-point adaptation methods: Full (full fine-tuning), Linear (fine-tuning the classification head), SAM-Adapter (Chen et al., 2023), LoRA (Hu et al., 2022); 2) standard QAT methods: LSQ (Esser et al., 2020) and NIPQ (Shin et al., 2023); 3) joint quantization and adaptation methods: PEQA (Kim et al., 2023), QA-LoRA (Xu et al., 2024), and QST (Zhang et al., 2024). Floating-point adaptation methods act as the upper-bound reference.

## 5.2 Semantic Segmentation Results

Table 1 presents the quantitative results on SAM-H. Our EQuA consistently achieves a superior trade-off between performance and efficiency across three segmentation domains: medical, natu-

Table 2: Results of adapting ViT-B on the VTAB-1K benchmark. Performance is reported in percentage (%). The best, second-best, and third-best results are marked in **bold**, underlined, and double underlined, respectively. The ✿ denotes $Param < 1M$, and 🍃 denotes $Mem < 2GB$.

| Method | Bit | Param (M) | Mem (GB) | Natural | | | | | | | Specialized | | | | Structured | | | | | | | | Averages | | | |
|---|---|---|---|---|---|---|---|---|---|---|---|---|---|---|---|---|---|---|---|---|---|---|---|---|---|---|---|
| | | | | Cifar100 | Caltech101 | DTD | Flower102 | Pets | SVHN | Sun397 | Camelyon | EuroSAT | Resisc45 | Retinopathy | Clevr-Count | Clevr-Dist | DMLab | KITTI-Dist | dSpr-Loc | dSpr-Ori | sNORB-Azim | sNORB-Ele | Avg Natural | Avg Specialized | Avg Structured | Average |
| Full | FP | 85.80 | 5.0 | 65.7 | 89.0 | 68.1 | 96.7 | 86.4 | 88.8 | 52.3 | 84.6 | 94.6 | 83.0 | 74.3 | 62.4 | 60.9 | 47.1 | 75.9 | 75.3 | 36.4 | 21.9 | 29.5 | 78.0 | 84.1 | 51.2 | 71.1 |
| Linear | FP | 0.00 | 0.6 | 64.4 | 85.0 | 63.2 | 97.0 | 86.3 | 36.6 | 51.0 | 78.5 | 87.5 | 68.5 | 74.0 | 34.3 | 30.6 | 33.2 | 55.4 | 12.5 | 20.0 | 9.6 | 19.2 | 69.1 | 77.1 | 26.9 | 57.6 |
| LoRA | FP | 0.69 | 4.3 | 74.9 | 89.1 | 73.3 | 99.2 | 91.8 | 83.9 | 57.1 | 85.7 | 96.1 | 87.2 | 74.2 | 72.1 | 60.6 | 49.4 | 76.8 | 72.0 | 54.2 | 26.4 | 35.2 | 81.3 | 85.8 | 55.8 | 74.3 |
| LSQ | W8A8 | 85.88 | 8.4 | 64.0 | 89.4 | 66.5 | 96.6 | 85.1 | 87.8 | 52.4 | 83.3 | 94.1 | 83.7 | 74.0 | 64.1 | 60.7 | 49.4 | 75.8 | 74.3 | 37.9 | 21.7 | 30.4 | 77.4 | 83.8 | 51.8 | 71.0 |
| NIPQ | W8A8 | 85.88 | 8.4 | 64.4 | 89.1 | 66.9 | 97.4 | 86.7 | 87.8 | 53.4 | 83.0 | 94.2 | 81.6 | 74.7 | 63.2 | 60.8 | 49.0 | 75.8 | **77.2** | 39.1 | 26.0 | 31.0 | 78.0 | 83.4 | 52.8 | 71.4 |
| QA-LoRA | W8A8 | ✿0.68 | 6.6 | **73.9** | 89.0 | 72.9 | 99.2 | **91.2** | 81.9 | **57.2** | 85.0 | **95.9** | **86.5** | 74.8 | 70.8 | 59.4 | 48.1 | 76.7 | 71.9 | **56.4** | 25.5 | 36.0 | **80.8** | **85.5** | 55.6 | **74.0** |
| PEQA | W8A8 | ✿0.08 | 6.7 | 67.9 | **90.0** | 69.5 | 98.6 | 88.4 | **88.9** | 52.4 | 87.1 | **95.9** | 85.6 | 75.1 | 76.6 | **61.8** | 51.1 | 79.5 | 77.0 | 51.6 | 31.4 | 38.9 | 79.3 | **85.9** | 58.5 | **74.6** |
| QST | W8A8 | ✿0.85 🍃1.2 | 62.0 | 88.9 | 71.0 | 99.2 | 89.7 | 70.3 | 56.0 | **87.7** | 95.2 | 80.6 | **75.7** | **79.0** | 61.1 | 47.9 | **79.7** | 74.0 | 29.8 | 30.6 | **44.5** | 76.7 | 84.8 | 55.8 | 72.4 |
| EQuA (Ours) | W8A8 | ✿0.64 🍃1.7 | 71.7 | 89.6 | **73.5** | 99.1 | 90.8 | 82.4 | 56.1 | 85.9 | 95.6 | 84.2 | 75.2 | 77.0 | 60.4 | 42.0 | 77.5 | 74.5 | 43.3 | 30.0 | 42.5 | 80.5 | 85.2 | 55.9 | 73.9 |
| LSQ | W4A4 | 85.88 | 8.4 | 59.3 | 86.9 | 65.3 | 91.8 | 82.1 | 81.6 | 44.7 | 79.6 | 94.1 | 80.2 | 72.4 | 58.4 | 59.5 | 45.8 | 74.9 | 74.3 | 33.0 | 11.5 | 29.5 | 73.1 | 81.6 | 48.4 | 67.7 |
| NIPQ | W4A4 | 85.88 | 8.4 | 59.3 | 87.1 | 65.9 | 93.6 | 78.8 | 85.7 | **48.8** | 82.5 | 93.2 | 80.5 | 72.8 | 60.3 | 60.6 | 40.3 | 74.7 | 75.2 | 36.7 | 18.6 | 30.0 | 74.2 | 82.3 | 49.6 | 68.7 |
| QA-LoRA | W4A4 | ✿0.68 | 6.6 | **68.4** | 88.8 | 71.2 | **99.0** | 89.8 | 83.2 | **52.8** | 84.3 | 95.5 | **85.5** | 71.0 | 70.3 | 59.2 | 42.8 | 76.1 | 71.0 | 51.0 | 25.0 | 35.6 | **79.0** | 84.1 | 54.5 | 72.5 |
| PEQA | W4A4 | ✿0.08 | 6.7 | 63.0 | 88.4 | 67.7 | 98.0 | 86.2 | **88.4** | 47.7 | **85.5** | **95.9** | 84.6 | 72.4 | 73.9 | 61.5 | **49.7** | 77.5 | **75.8** | 51.5 | 30.1 | 35.8 | 77.1 | **84.6** | **57.0** | **72.9** |
| QST | W4A4 | ✿0.85 🍃1.2 | 47.8 | 87.8 | 67.6 | 98.1 | 86.6 | 67.9 | 48.5 | 84.1 | 95.0 | 80.0 | **75.4** | **77.2** | 60.1 | 46.6 | 76.9 | 73.3 | 28.3 | 27.7 | **43.4** | 72.0 | 83.7 | 54.2 | 70.0 |
| EQuA (Ours) | W4A4 | ✿0.64 🍃1.7 | 63.6 | **88.8** | **72.3** | 98.6 | **88.3** | 80.5 | 51.8 | 85.3 | 95.1 | 81.4 | 73.6 | 74.1 | 60.0 | 40.0 | 76.1 | 72.7 | 41.4 | 29.7 | 40.0 | 77.7 | 83.9 | 54.3 | 72.0 |

Table 3: Ablation on the each components.

| Components | Param↓ (M) | Train↓ (hour/epoch) | Mem↓ (GB) | Kvasir $S_\alpha \uparrow$ | $E_\phi \uparrow$ |
|---|---|---|---|---|---|
| Base | 641.46 | 0.60 | 72.4 | 0.895 | 0.931 |
| +SSA* | 29.90 | 0.57 | 67.6 | 0.918 | 0.941 |
| +SSA | 29.90 | 0.32 | 18.6 | 0.881 | 0.913 |
| +SSA & GC | 29.90 | 0.50 | 18.6 | 0.910 | 0.932 |
| +SSA & GC & LoRA | 7.67 | 0.50 | 18.5 | 0.906 | 0.926 |
| +SSA & GC & LoRA & BAQF | 7.67 | 0.52 | 18.5 | 0.915 | 0.947 |

Table 4: Ablation on hyper-parameters $D_s$ and $r$.

| $D_s$ | $r$ | Param↓ (M) | Mem↓ (GB) | CVC-T $S_\alpha \uparrow$ | $E_\phi \uparrow$ | Kvasir $S_\alpha \uparrow$ | $E_\phi \uparrow$ |
|---|---|---|---|---|---|---|---|
| 32 | 4 | 7.66 | 18.3 | 0.915 | 0.936 | 0.906 | 0.933 |
| 64 | 4 | 7.67 | 18.5 | 0.922 | 0.938 | 0.915 | **0.947** |
| 128 | 4 | 7.69 | 18.7 | 0.927 | 0.947 | **0.920** | 0.943 |
| 64 | 2 | 6.33 | 18.5 | 0.918 | 0.935 | 0.906 | 0.932 |
| 64 | 8 | 10.35 | 18.5 | **0.930** | **0.954** | 0.919 | 0.944 |

ral, and agricultural. Standard QAT methods fine-tune all parameters and incur substantial memory overheads (over 70 GB), making them inefficient in both parameter and memory. While QA-LoRA and PEQA reduce the number of trainable parameters, their memory consumption remains above 60 GB. In contrast, our EQuA achieves comparable performance with significantly lower memory cost. For example, on the CVC-T dataset under the W4A4 setting, EQuA achieves an $S_\alpha$ of 0.922, closely matching the best result of PEQA (0.927) while reducing memory consumption from 69.4GB to 18.5 GB (a 73.3% reduction), enabling the quantization-aware adaptation of SAM-H on a single RTX 3090 GPU. Compared to the memory-efficient QST, our EQuA achieves better performance. For example, on the Leaf dataset under the W4A4 setting, EQuA outperforms QST by 0.053 mIoU and 0.044 mDice (0.709/0.816 vs. 0.656/0.772). As shown in Fig. 3, we also visualize the comparison results. The results demonstrate our EQuA can also achieve superior qualitative performance. In summary, our EQuA offers a strong balance between performance and efficiency. More quantitative and qualitative results are provided in Appendix I.

## 5.3 IMAGE CLASSIFICATION RESULTS

In Table 2, we report the comparison results on ViT-B. The results further confirm that our EQuA continues the advantage in balancing performance and efficiency across diverse downstream visual tasks on ViT-B. For example, our EQuA requires only 1.7 GB of training memory, achieving a reduction of over 70% compared to PEQA and QA-LoRA, and over 75% compared to LSQ and NIPQ. Our EQuA achieves comparable or superior performance on all 19 datasets. Under the W4A4 setting, EQuA attains an average Top-1 accuracy of 72.0%, significantly outperforming QST (70.0%) and NIPQ (68.7%), and closely matching PEQA (72.9%) and QA-LoRA (72.5%).

## 6 ABLATION AND ANALYSIS

### 6.1 EFFECTIVENESS OF EACH COMPONENT

In order to reveal the effectiveness of each component of our EQuA, we report the ablation results under the W4A4 setting in Table 3. LSQ, the first row in Table 3, serves as the reference baseline without any additional components. *SSA** denotes fine-tune only the SSA branch, but the MIB

Table 5: Efficiency analysis during adaptation (left) and deployment (right), which are obtained by setting batch size to 1/2/4, respectively. *Speedup*: inference acceleration relative to the original floating-point model. *OOM*: out-of-memory. Models are trained on a single A100 GPU (80 GB). The input image resolution is $1024 \times 1024 \times 3$.

| Method | Memory↓ (GB) | Train↓ (hour/epoch) | Storage↓ (GB) | Latency↓ (sec/batch) | Speedup↑ |
|---|---|---|---|---|---|
| Full | 28.6/49.6/OOM | 0.57/0.55/OOM | 2.39 | 0.75/1.44/2.94 | 1.00×/1.00×/1.00× |
| SAM-Adapter | 18.5/34.7/66.7 | **0.37/0.35/0.34** | 2.39 | 0.88/1.67/3.38 | 0.85×/0.86×/0.86× |
| PEQA | 36.7/69.4/OOM | 0.63/0.58/OOM | **0.34** | **0.54/1.06/2.14** | **1.38×/1.36×/1.37×** |
| QA-LoRA | 35.9/68.9/OOM | 0.58/0.55/OOM | **0.34** | **0.54/1.06/2.14** | **1.38×/1.36×/1.37×** |
| QST | 10.9/21.6/42.0 | 0.41/0.37/0.35 | 0.35 | 0.63/1.24/2.49 | 1.18×/1.16×/1.18× |
| EQuA (Ours) | **10.7/18.5/33.9** | 0.55/0.52/0.50 | **0.34** | **0.54/1.06/2.14** | **1.38×/1.36×/1.37×** |

branch is included in the computation graph. *SSA* denotes fine-tune only the SSA branch, but the MIB branch is excluded from the computation graph. *GC* denotes the gradient checkpointing. *LoRA* represents using LoRA in Eq. 10. As shown in Table 3, adopting SSA significantly reduces trainable parameters, training time, and memory usage. GC intergrates partial quantization-aware information from the MIB branch into the SSA branch, further facilitating convergence and yielding a significant performance gain (0.910 vs. 0.881 on $S_\alpha$) with negligible memory overhead. Although the training time increases moderately, it is still lower than that of LSQ (0.50 vs. 0.60). Incorporating LoRA maintains performance and reduces trainable parameters to just 1.2% of LSQ, which can achieve efficient model switching across different downstream tasks. Finally, BAQF further improve performance without additional memory cost, by alleviating quantization errors and memory consumption from activation quantizers.

## 6.2 Ablation on Hyper-Parameters

In order to reveal the impact of different hyper-parameters of our EQuA, we report the ablation study on the SSA dimension $D_s$ in Eq. 7 and the LoRA rank $r$ in Eq. 10.

**SSA dimension** $D_s$**.** We set $r$ to 4 when studying the effect of $D_s$. As listed in Table 4, increasing $D_s$ from 32 to 128 results in negligible growth trainable parameters and only a slight increase in training memory consumption during adaptation. A larger performance gain is observed when increasing $D_s$ from 32 to 64 (*e.g.*, a 0.014 improvement on $E_\phi$ in the Kvasir dataset), while the performance are comparable between $D_s = 64$ and $D_s = 128$ . Thus, we adopt $D_s = 64$ as the default configuration.

**LoRA rank** $r$**.** In this set of experiments, we set $D_s$ to 64 and change $r$. As shown in Table 4, increasing $r$ from 2 to 8 significantly increases trainable parameters (from 6.33 M to 10.35 M), while training memory consumption keeps unchanged. However, the corresponding performance improvements become marginal beyond $r = 4$, *e.g.*, 0.915 *vs.* 0.919 on $S_\alpha$ in the Kvasir dataset. This suggests that $r = 4$ offers a good balance between parameter efficiency and performance.

## 6.3 Analysis of Efficiency

Table 5 summarizes the efficiency analysis on SAM-H. PEQA, QA-LoRA, QST, and our EQuA are under the W4A4 quantization setting, while Full and SAM-Adapter remain in floating-point. *Memory* and *Train* reflect adaptation efficiency, which are evaluated on the polyp segmentation training set. *Storage*, *Latency*, and *Speedup* reflect deployment efficiency. We adopt the CUDA kernels (Cho et al., 2025) based on the CUTLASS library for 4-bit inference acceleration to estimate *Latency* and *Speedup*. This library is developed by NVIDIA and enable a practical evaluation of real-world hardware latency. As shown in Table 5, our EQuA achieves a significant advantage in training efficiency, reducing 73.3% training memory and 10.3% training time compared to PEQA (batch size 2). At deployment, EQuA preserves all the benefits of standard 4-bit quantization, achieving a 1.38× inference speedup with a batch size of 1. Although QST shows similar memory savings, its speedup is lower than that of standard 4-bit quantization due to the extra overhead of the additional adapters. In contrast, our EQuA achieves both training and deployment efficiency simultaneously.

## 7 Conclusion

We propose EQuA, an efficient quantization-aware adaptation framework for visual foundation models. To reduce the high memory overhead of joint quantization and adaptation, we introduce an

SSA branch to enable gradient updates without full-model backpropagation and SAQF and BAQF strategies to preserve memory efficiency and minimize quantization errors. At deployment, SSA is merged back into the backbone, incurring no extra computation costs. Extensive experiments validate that EQuA achieves a optimal trade-off between performance and efficiency, facilitating practical both quantization-aware adaptation and deployment under resource constraints.

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

## A  MATHEMATICAL DETAILS OF WEIGHT IMPORTANCE

Our purpose of evaluating weight importance is similar to model pruning techniques: determining which weights dominate the model performance. Then, we select important weight channels to construct the SSA branch and perform fine-tuning. Based on this idea, we follow model pruning metric of Wanda (Sun et al., 2024) to estimate weight importance in our work. Here, we provide the mathematical derivation of the weight importance from the perspective of model pruning with reference to OBS (Hassibi et al., 1993).

Consider a linear layer with input matrix $X \in \mathbb{R}^{BN \times D_{in}}$ and weight matrix $W \in \mathbb{R}^{D_{in} \times D_{out}}$. Given target outputs $Y$, we define the local reconstruction loss:

$$\mathcal{L}(W) = \frac{1}{2} \left\| WX - Y \right\|_F^2. \tag{12}$$

Suppose we aim to prune (set to zero) a specific weight $W_{ij}$ while allowing all other weights to readjust so that the increase in loss is is minimized. For brevity, we flatten $W$ into a vector $w$. Since the pre-trained foundation model has converged to a minimum, the gradients can be considered close to zero, around the optimum $w$ (where $\nabla \mathcal{L}(w) = 0$). The increase in loss can be obtained using the second-order Taylor expansion:

$$\begin{aligned} \Delta \mathcal{L} = \mathcal{L}(w + \Delta w) - \mathcal{L}(w) &\approx \nabla \mathcal{L}(w) \Delta w + \tfrac{1}{2} \Delta w^\top H \Delta w \\ &= \tfrac{1}{2} \Delta w^\top H \Delta w, \end{aligned} \tag{13}$$

where $H = \nabla^2 \mathcal{L}(w)$ is the Hessian matrix. The following constraint means pruning weight $w_k$:

$$e_k^\top (w + \Delta w) = 0, \tag{14}$$

where $e_k$ is the standard basis vector selecting index $k$. We minimize the quadratic form $\frac{1}{2} \Delta w^\top H \Delta w$ subject to this linear constraint (Eq. 14). The Lagrangian is calculated as:

$$\mathcal{J}(\Delta w, \lambda) = \tfrac{1}{2} \Delta w^\top H \Delta w + \lambda \, e_k^\top (w + \Delta w). \tag{15}$$

Differentiating $\mathcal{J}(\Delta w, \lambda)$ with respect to $\Delta w$ and setting it to 0, we can obtain $\Delta w = -\lambda H^{-1} e_k$. Substituting this into the Eq. 14 yields:

$$\lambda = \frac{w_k}{(H^{-1})_{kk}}. \tag{16}$$

The minimum loss increase is therefore:

$$\Delta \mathcal{L}_k = \tfrac{1}{2} \Delta w^\top H \Delta w = \frac{1}{2} \frac{w_k^2}{(H^{-1})_{kk}}. \tag{17}$$

The Eq. 17 shows the loss increases in direct proportion to $\frac{w_k^2}{(H^{-1})_{kk}}$, which can reflect the weight importance. Hence the weight importance of $w_k$ is estimated as:

$$\boxed{\frac{w_k^2}{(H^{-1})_{kk}}}. \tag{18}$$

For the squared reconstruction loss (Eq. 12), the Hessian with respect to $w$ factorizes as $H = (X^\top X) \otimes I_m$, where $\otimes$ is the Kronecker product. The entry of $H^{-1}$ corresponding to weight $w_{ij}$ depends only on the $j$–th input dimension:

$$(H^{-1})_{kk} = \left[ (X^\top X)^{-1} \right]_{jj} = \mathrm{diag}((X^\top X)^{-1})_j, \tag{19}$$

where $\mathrm{diag}(\cdot)$ extracts the diagonal of a matrix. Since the Eq. 19 incurs an expensive calculation, Sun et al. (2024) simplify it with a proxy calculation: $\mathrm{diag}((X^\top X)^{-1}) \approx \mathrm{diag}(X^\top X)^{-1}$. This proxy eliminates the need to explicitly compute matrix inverses, thereby greatly reducing the computation burden. Finally, the importance of the weight at index $(i, j)$ can be quickly estimated as:

$$\boxed{S_{ij} = \sqrt{\left[ \frac{|W|^2}{\mathrm{diag}(X^\top X)^{-1}} \right]_{ij}} = \sqrt{(|W_{ij}| \cdot \|X_i\|_2)^2} = |W_{ij}| \cdot \|X_i\|_2}, \tag{20}$$

where $|W|^2$ is the element-wise square of the weight $W$, the division is element-wise. Therefore, based on Eq. 20, we estimate the weight importance of each channel by accumulating the weight importance in this channel:

$$\boxed{M_i = \sum_{j=1}^{D_{out}} S_{ij}}, \tag{21}$$

where $M_i$ denotes the weight importance of $i$-th input channel, as demonstrated in Eq. 8.

## B  MORE ANALYSIS ON SELECTION OF SSA CHANNELS

Selecting important weight channels is crucial for constructing the SSA branch. In our work, we perform stochastic selection of SSA channels based on the computed weight importance probabilities in Eq. 8, whose detailed mathematical formula is described as follows:

$$\text{Strategy A: } \mathcal{I}_{ssa} = Sampling(D_s; P^M), \tag{22}$$

where $Sampling(D_s; P^M)$ represents stochastically sampling $D_s$ weight channels in the original backbone weight channels from the probability distribution $P^M$ without replacement. **We adopt Strategy A as our design**. In Table 7, we report the time and memory cost to perform Strategy A and show a performance-memory trade-off on SAM-H (W4A4) across different channel splitting ratios in Fig. 4c.

To evaluate the effectiveness of our selection strategy, *i.e.*, Strategy A, we make comparisons with some other optional selection strategies as follows:

$$\text{Strategy B: } \mathcal{I}_{ssa} = random(D_s),$$

$$\text{Strategy C: } \mathcal{I}_{ssa} = \arg\max_{D_s} \mathcal{M},$$

$$\text{Strategy D: } \mathcal{I}_{ssa} = \arg\min_{D_s} \mathcal{M},$$

$$\text{Strategy E: } \mathcal{I}_{ssa} = \arg\max_{D_s} \sum_{j=1}^{D_{out}} |W_{:,j}|, \tag{23}$$

$$\text{Strategy F: } \mathcal{I}_{ssa} = \arg\min_{D_s} \sum_{j=1}^{D_{out}} |W_{:,j}|,$$

$$\text{Strategy G: } \mathcal{I}_{ssa} = \arg\max_{D_s} \frac{\partial L}{\partial W},$$

where $random(D_s)$ denotes sampling $D_s$ weight channels without replacement from a uniform distribution. $\mathcal{M} = [M_1, M_2, ..., M_{D_{in}}]$ and $M_i$ denotes the weight importance of $i$-th input channel (Eq. 8). Strategies C and D select the top-$D_s$ and bottom-$D_s$ input channels of weights based on weight importance to construct the SSA channels, respectively. Strategies E and F select the input channels of weights with the the top-$D_s$ and bottom-$D_s$ absolute weight magnitudes, respectively. In table 6, we make comparisons between these strategies. The results demonstrate our desgin, *i.e.*, Strategy A, deliver superior performance for selecting SSA channels. Notably, Strategy A generally outperforms Strategy C in Table 6. Because Strategy C fine-tunes only the weights with the largest scaled magnitudes, these weights, although few in number, dominate model performance and their adjustment is close to fine-tuning the entire visual foundation model. However, downstream datasets are much smaller than the large-scale pre-training datasets, so updating only the dominant weights on such limited data increases the risk of overfitting and limits generalization. In contrast, the random selection used in Strategy A helps mitigate this problem. **It is the reason why we adopt the stochastic Strategy A rather than deterministic Strategy C**. As shown in Fig. 4b, Strategy C shows an overfitting trend on a small dataset. Moreover, Strategy C outperforms Strategy E, primarily because Strategy C incorporates input information from downstream tasks as guidance during the selection process.

To further evaluate the stability of Strategy A, we trained SAM-H three times using different random seeds and report the corresponding mean and standard deviation. The results, summarized in Table 8, demonstrate that Strategy A exhibits stability and robustness.

## C  ABLATION ON BLOCK-WISE FINE-TUNING FREQUENCY

Table 9 reports the impact of block-wise fine-tuning frequency $K$ of our Block-wise Activation Quantizer Fine-tuning (BAQF) strategy, where $K = 0$ represents we do not perform BAQF.

## D  MORE ANALYSIS ON QST IN VISUAL FOUNDATION MODELS

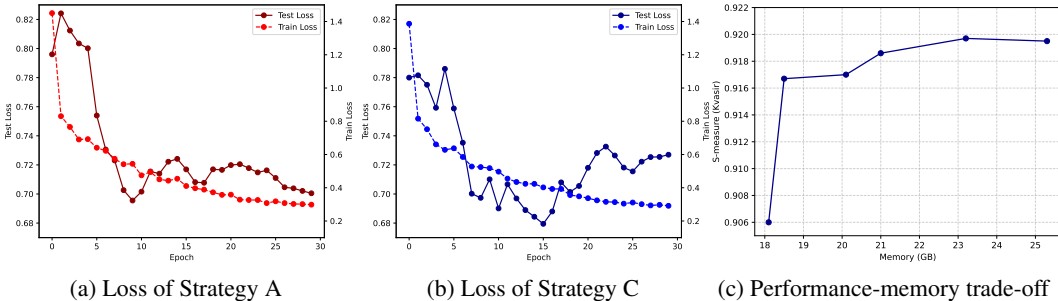

| (a) Loss of Strategy A | (b) Loss of Strategy C | (c) Performance-memory trade-off |
| --- | --- | --- |

Figure 4: The results are tested on SAM-H (W4). (a) and (b) are the loss of Strategy A and Strategy C on a small training set and a testing set (CVC-T dataset (Vázquez et al., 2016)). The small training set are sampled from the Polyp Segmentation training set. (c) Performance–memory trade-off across channel splitting ratios ($\frac{D_s}{D}$) with our proposed EQuA. Channel splitting ratios are set to 0.03/0.1/0.3/0.5/0.7/1.0, respectively

Table 6: Different selection strategies of SSA channels under W4A4 setting.

| Strategy | Mem ↓ (GB) | CVC-T $S_\alpha$ ↑ | CVC-T $E_\phi$ ↑ | Kvasir $S_\alpha$ ↑ | Kvasir $E_\phi$ ↑ |
| --- | --- | --- | --- | --- | --- |
| A | 9.6 | **0.922** | **0.938** | 0.915 | **0.947** |
| B | 0.0 | 0.913 | 0.922 | 0.905 | 0.927 |
| C | 9.6 | 0.921 | 0.934 | **0.919** | 0.943 |
| D | 9.6 | 0.909 | 0.906 | 0.896 | 0.920 |
| E | 0.0 | 0.916 | 0.921 | 0.908 | 0.935 |
| F | 0.0 | 0.909 | 0.908 | 0.900 | 0.925 |
| G | 57.2 | 0.924 | 0.936 | 0.917 | 0.949 |

Table 7: The time and memory cost to perform Strategy A on SAM-H.

| Batch size | Time | Memory | Training Memory |
| --- | --- | --- | --- |
| 1 | 13.9 sec | 6.1 GB | 10.7 GB |
| 2 | 20.6 sec | 9.6 GB | 18.5 GB |
| 4 | 33.2 sec | 16.4 GB | 33.9 GB |
| 8 | 57.6 sec | 30.2 GB | 65.0 GB |

Here, we present an analysis of the dilemma faced by QST (Zhang et al., 2024) in visual foundation models. The models on vision tasks basically adopt both weight quantization and activation quantization to achieve efficient integer-arithmetic-only inference for a lower inference latency (Cho et al., 2025; Jacob et al., 2018). Activation quantization is often the main focus of quantization studies on vision tasks, as the performance of vision models is highly sensitive to it (Li et al., 2023; Lv et al., 2024). Nevertheless, the vanilla QST quantizes the backbone without considering optimizing activation quantizers. As shown in Fig. 5, quantization errors from quantized block accumulate in the deep backbone, producing low-quality intermediate features that are injected into the side adapter, ultimately resulting in performance degradation on vision tasks. As shown in Table 10, although the 4-bit weights of backbone lower the performance compared to 8-bit weights, the performance decreases further at a larger margin when the activation bit change from W4A8 to W4A4, which verifies the affect of activation quantization on QST on vision tasks.

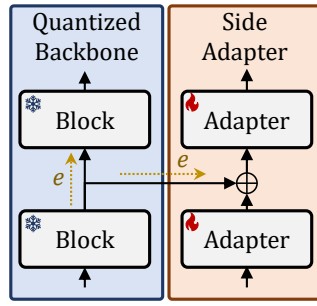

Figure 5: Quantization error flow in QST. The yellow $e$ represents quantization errors from the quantized backbone.

# E    DETAILS OF GRADIENT CHECKPOINTING

Here, we explain the role of gradient checkpointing in our SAQF by taking optimizing $MHSA_{ssa}$ module as an example. For simplicity, we rewrite Eq. 3 into a simplified form, which does not affect the conclusion:

$$Y_{l-1} = MHSA(F_{l-1}),$$
$$F_l = MLP(Y_{l-1}),$$

(24)

Table 8: Stability results of Strategy A based on three runs of SAM-H fine-tuning with different random seeds under W4A4 setting.

| SAM-H | CVC-T | | Kvasir | | ETIS | |
|---|---|---|---|---|---|---|
| | $S_\alpha \uparrow$ | $E_\phi \uparrow$ | $S_\alpha \uparrow$ | $E_\phi \uparrow$ | $S_\alpha \uparrow$ | $E_\phi \uparrow$ |
| mean±std | $0.924_{\pm 0.005}$ | $0.940_{\pm 0.004}$ | $0.917_{\pm 0.002}$ | $0.945_{\pm 0.002}$ | $0.795_{\pm 0.003}$ | $0.833_{\pm 0.003}$ |

Table 9: Ablation on the frequency $K$ of BAQF.

| $K$ | Leaf | |
|---|---|---|
| | mIoU↑ | mDice↑ |
| 0 | 0.699 | 0.808 |
| 5 | 0.708 | **0.816** |
| 10 | **0.709** | **0.816** |
| 15 | 0.707 | 0.813 |

Table 10: Analysis on QST.

| Backbone Bit | Leaf | |
|---|---|---|
| | mIoU↑ | mDice↑ |
| W8A8 | 0.678 | 0.792 |
| W4A8 | 0.670 | 0.786 |
| W4A4 | 0.656 | 0.772 |

According to Eq. 24, the update of $MHSA_{ssa}$ can be formulated as follows:

$$W_{ssa}^{MHSA} \leftarrow W_{ssa}^{MHSA} - \eta \cdot \nabla_{W_{ssa}^{MHSA}} \mathcal{L},$$

$$\nabla_{W_{ssa}^{MHSA}} \mathcal{L} = \frac{\partial Y_{l-1}}{\partial W_{ssa}^{MHSA}} \cdot \frac{\partial F_l}{\partial Y_{l-1}} \cdot \frac{\partial \mathcal{L}}{\partial F_l}, \tag{25}$$

where $W_{ssa}^{MHSA}$ represents the trainable LoRA parameters ($A_r$ and $B_r$) and scaling factors ($s_{w_{ssa}}$) of weight quantizers in $MHSA_{ssa}$. The gradient $\frac{\partial F_l}{\partial Y_{l-1}}$ provides the information in the $MLP$ module. Since we split the $MLP$ module into the frozen $MLP_{mib}$ and the trainable $MLP_{ssa}$ according to Eq. 7, the gradient $\frac{\partial F_l}{\partial Y_{l-1}}$ can be calculated as:

$$\frac{\partial F_l}{\partial Y_{l-1}} = \frac{\partial MLP_{ssa}(Y_{l-1})}{\partial Y_{l-1}}. \tag{26}$$

The Eq. 26 only provides the gradient information in $MLP_{ssa}$, without considering the quantization error accumulated in $MLP_{mib}$, which degrades the performance. Therefore, we introduce gradient checkpointing in this position as follows:

$$\frac{\partial F_l}{\partial Y_{l-1}} = \frac{\partial MLP_{ssa}(Y_{l-1})}{\partial Y_{l-1}} + GC(Y_{l-1}), \tag{27}$$

where $GC(\cdot)$ represents gradient checkpointing, which recomputes the gradient $\frac{\partial MLP_{mib}(Y_{l-1})}{\partial Y_{l-1}}$ on the fly. Obviously, the gradient $\frac{\partial F_l}{\partial Y_{l-1}}$ in Eq. 27 is integrated with quantization-aware information in $MLP_{mib}$, which enables $MHSA_{ssa}$ to compensate for the performance degradation. Moreover, this design also avoids constructing computation graph for MIB branch when computing gradient $\frac{\partial MLP_{mib}(Y_{l-1})}{\partial Y_{l-1}}$, which would otherwise incur substantial additional memory consumption. Specifically, we compute $GC(X_{mib}^V)$ and $GC(X_{mib}^{Lin1})$, where $X_{mib}^V$ and $X_{mib}^{Lin1}$ denote the input activations of $V$ layer and $Lin1$ layer in the MIB branch, respectively.

Thanks to the absence of learnable parameters in our MIB branch, the $GC(\cdot)$ in our design can recompute the gradients of the $X_{mib}^V$ and $X_{mib}^{Lin1}$ using a one-step analytical computation. This is different from the regular gradient checkpointing (Chen et al., 2016), which requires an extra forward propagation step to recompute the intermediate activations and then performs a backward propagation step to recompute the gradients. The one-step computation of our $GC(\cdot)$ is as follows:

$$GC(X_{mib}^V): \frac{\partial L}{\partial X_{mib}^V} = (A^T((\frac{\partial L}{\partial Y_{mib}^{Proj}} W_{mib}^{Proj,T}) \odot \mathbb{1}_{mib}^{Proj}) \odot \mathbb{1}_{mib}^V) W_{mib}^{V,T},$$

$$GC(X_{mib}^{Lin1}): \frac{\partial L}{\partial X_{mib}^{Lin1}} = (\frac{\partial L}{\partial Y_{mib}^{Lin2}} W_{mib}^{Lin2,T} \odot \mathbb{1}_{mib}^{Lin2}) \phi(\hat{X}_{mib}^{Lin1} W_{mib}^{Lin1}) W_{mib}^{Lin1,T} \odot \mathbb{1}_{mib}^{Lin1}, \tag{28}$$

where $\odot$ is the element-wise multiplication. $W_{mib}^{Proj,T}$ denotes the transpose of $W_{mib}^{Proj}$, and the other notations follow analogously. $\hat{X}_{mib}^{Lin1}$ is the de-quantized $X_{mib}^{Lin1}$. $Y_{mib}^{Proj}$ and $Y_{mib}^{Lin2}$ denote the output of $Proj$ layer and $Lin2$ layer in the MIB branch, respectively. $\phi(\cdot)$ represents the analytical gradients of GeLU function. $\mathbb{1}_{mib}^V$, $\mathbb{1}_{mib}^{Proj}$, $\mathbb{1}_{mib}^{Lin1}$, and $\mathbb{1}_{mib}^{Lin2}$ are boolean mask tensors, which are computed from activation quantizers of $X_{mib}^V$, $X_{mib}^{Proj}$, $X_{mib}^{Lin1}$, and $X_{mib}^{Lin2}$, respectively. For

Table 11: Experimental Setup. *Iteration* represents the number of times each block is fine-tuned in each round of BAQF.

| Task | $lr_{backbone}$ | $lr_{decoder}$ | Epoch | Iteration |
|---|---|---|---|---|
| Camouflaged Object Detection | $1 \times 10^{-4}$ | $1 \times 10^{-4}$ | 30 | 100 |
| Polyp Segmentation | $1 \times 10^{-4}$ | $1 \times 10^{-4}$ | 30 | 100 |
| Leaf Disease Segmentation | $3 \times 10^{-4}$ | $3 \times 10^{-4}$ | 20 | 100 |
| Image Classification on VTAB | $5 \times 10^{-4}$ | $5 \times 10^{-3}$ | 100 | 100 |

Table 12: Semantic segmentation results of SAM-H on camouflaged object detection datasets.

| Method | Bit | Param↓ | Mem↓ | CAMO $S_\alpha \uparrow$ | $E_\phi \uparrow$ | $F_\beta^\omega \uparrow$ | MAE↓ | COD10K $S_\alpha \uparrow$ | $E_\phi \uparrow$ | $F_\beta^\omega \uparrow$ | MAE↓ | NC4K $S_\alpha \uparrow$ | $E_\phi \uparrow$ | $F_\beta^\omega \uparrow$ | MAE↓ |
|---|---|---|---|---|---|---|---|---|---|---|---|---|---|---|---|
| Full | FP | 641.09 M | 49.6 GB | 0.805 | 0.860 | 0.734 | 0.077 | 0.837 | 0.892 | 0.736 | 0.034 | 0.856 | 0.900 | 0.794 | 0.047 |
| LoRA | FP | 8.03 M | 45.4 GB | 0.889 | 0.933 | 0.863 | 0.041 | 0.920 | 0.959 | 0.881 | 0.015 | 0.913 | 0.947 | 0.886 | 0.027 |
| LSQ | W8A8 | 641.46 M | 72.4 GB | 0.808 | 0.864 | 0.746 | 0.074 | 0.843 | 0.900 | 0.747 | 0.032 | 0.854 | 0.900 | 0.796 | 0.047 |
| NIPQ | W8A8 | 641.46 M | 72.4 GB | 0.805 | 0.859 | 0.745 | 0.076 | 0.841 | 0.898 | 0.747 | 0.032 | 0.855 | 0.900 | 0.799 | 0.047 |
| QA-LoRA | W8A8 | 7.13 M | 68.9 GB | 0.885 | 0.928 | 0.851 | 0.045 | 0.919 | 0.959 | 0.876 | 0.015 | 0.912 | 0.943 | 0.880 | 0.028 |
| PEQA | W8A8 | 4.43 M | 69.4 GB | 0.877 | 0.923 | 0.843 | 0.047 | 0.909 | 0.951 | 0.860 | 0.018 | 0.901 | 0.937 | 0.866 | 0.032 |
| QST | W8A8 | 8.30 M | 21.6 GB | 0.839 | 0.884 | 0.781 | 0.069 | 0.885 | 0.930 | 0.814 | 0.024 | 0.880 | 0.915 | 0.823 | 0.041 |
| EQuA (Ours) | W8A8 | 7.67 M | 18.5 GB | 0.863 | 0.900 | 0.813 | 0.058 | 0.902 | 0.942 | 0.844 | 0.020 | 0.897 | 0.928 | 0.852 | 0.034 |
| LSQ | W4A4 | 641.46 M | 72.4 GB | 0.760 | 0.804 | 0.666 | 0.092 | 0.806 | 0.867 | 0.686 | 0.039 | 0.829 | 0.873 | 0.755 | 0.056 |
| NIPQ | W4A4 | 641.46 M | 72.4 GB | 0.762 | 0.808 | 0.659 | 0.094 | 0.818 | 0.875 | 0.689 | 0.037 | 0.842 | 0.886 | 0.763 | 0.052 |
| QA-LoRA | W4A4 | 7.13 M | 68.9 GB | 0.858 | 0.903 | 0.811 | 0.056 | 0.903 | 0.945 | 0.846 | 0.019 | 0.903 | 0.935 | 0.863 | 0.032 |
| PEQA | W4A4 | 4.43 M | 69.4 GB | 0.855 | 0.899 | 0.809 | 0.057 | 0.892 | 0.937 | 0.830 | 0.022 | 0.894 | 0.929 | 0.851 | 0.035 |
| QST | W4A4 | 8.30 M | 21.6 GB | 0.788 | 0.836 | 0.694 | 0.089 | 0.833 | 0.884 | 0.718 | 0.037 | 0.842 | 0.882 | 0.760 | 0.055 |
| EQuA (Ours) | W4A4 | 7.67 M | 18.5 GB | 0.833 | 0.878 | 0.766 | 0.073 | 0.873 | 0.920 | 0.790 | 0.026 | 0.875 | 0.909 | 0.810 | 0.043 |

example, the meaning of $\mathbb{1}_{mib}^V$ is below, and the other notations follow analogously:

$$[\mathbb{1}_{mib}^V]_{ij} = \begin{cases} 1, & \text{if } 0 \leq [Quant(X_{mib}^V)]_{ij} \leq 2^b - 1 \\ 0, & \text{otherwise} \end{cases} \qquad X_{mib}^V \in \mathbb{R}^{BN \times D_{in}}. \qquad (29)$$

**Note that the activation quantizers and input activations of $V$ and $Lin1$ layers in MIB branch are also the same activation quantizers and input activations of $V$ and $Lin1$ layers in SSA branch**, *i.e.*, $X_{mib}^V = X_{ssa}^V$, $X_{mib}^{Lin1} = X_{ssa}^{Lin1}$, $\hat{X}_{mib}^{Lin1} = \hat{X}_{ssa}^{Lin1}$, $\mathbb{1}_{mib}^V = \mathbb{1}_{ssa}^V$, and $\mathbb{1}_{mib}^{Lin1} = \mathbb{1}_{ssa}^{Lin1}$. Therefore, all tensors in Eq. 28, except for $\mathbb{1}_{mib}^{Proj}$ and $\mathbb{1}_{mib}^{Lin2}$, are already shared by the SSA branch and do not need to be recomputed. We manually cache $\mathbb{1}_{mib}^{Proj}$ and $\mathbb{1}_{mib}^{Lin2}$ during the forward pass before loss computation.

## F  DETAILS OF EXPERIMENT SETTINGS

All experiments are conducted using PyTorch. The learning rates for different tasks are summarized in Table 11. $lr_{backbone}$ represents the learning rate of fine-tuning backbone, and $lr_{decoder}$ represents the learning rate of fine-tuning decoder of SAM-H or classification head of ViT-B. For the BAQF strategy, we set the batch size to 2 for SAM-H and 32 for ViT-B, and perform only 100 iterations per round for each block. We further report the ablation study on backbone learning rate in Table 14 and the number of iteration of BAQF in Table 15 for SAM-H (W4A4).

## G  MORE DETAILS OF EQUA PROCESSING PIPELINE

The processing pipeline of our EQuA is shown in Algorithm 1, where $B_r$ and $A_r$ are the LoRA matrices (Eq. 10), $s_{w_{ssa}}$ is the scaling factor of weight quantizers in SSA (Eq. 10), $W_{ssa}$ represents the weights of SSA, and $s_a$ represents the scaling factor of activation quantizers.

## H  DETAILS OF DATASETS AND METRICS

To jointly adapt and quantize SAM for downstream tasks, we follow prior works (Zhong et al., 2024) and include three downstream tasks: polyp segmentation, camouflaged object detection, and leaf disease segmentation. We also use six metrics for evaluation: mean Dice score (mDice), mean IoU score (mIoU), mean absolute error (MAE), weighted F-measure ($F_\beta^\omega$), E-measure ($E_\phi$), and S-measure ($S_\alpha$).

Table 13: Semantic segmentation results of SAM-H on polyp segmentation datasets. We omit $Param$ and $Mem$ for brevity.

| Method | Bit | CVC-T | | | | Kvasir | | | | ETIS | | | | CVC-612 | | | | CVC-ColonDB | | | |
|---|---|---|---|---|---|---|---|---|---|---|---|---|---|---|---|---|---|---|---|---|---|
| | | $S_\alpha\uparrow$ | $E_\phi\uparrow$ | $F_\beta^\omega\uparrow$ | MAE↓ | $S_\alpha\uparrow$ | $E_\phi\uparrow$ | $F_\beta^\omega\uparrow$ | MAE↓ | $S_\alpha\uparrow$ | $E_\phi\uparrow$ | $F_\beta^\omega\uparrow$ | MAE↓ | $S_\alpha\uparrow$ | $E_\phi\uparrow$ | $F_\beta^\omega\uparrow$ | MAE↓ | $S_\alpha\uparrow$ | $E_\phi\uparrow$ | $F_\beta^\omega\uparrow$ | MAE↓ |
| Full | FP | 0.917 | 0.940 | 0.838 | 0.011 | 0.917 | 0.954 | 0.898 | 0.027 | 0.815 | 0.833 | 0.652 | 0.030 | 0.933 | 0.971 | 0.904 | 0.014 | 0.833 | 0.877 | 0.734 | 0.043 |
| LoRA | FP | 0.937 | 0.963 | 0.879 | 0.011 | 0.936 | 0.962 | 0.915 | 0.022 | 0.845 | 0.857 | 0.706 | 0.033 | 0.932 | 0.963 | 0.893 | 0.016 | 0.853 | 0.878 | 0.758 | 0.048 |
| LSQ | W8A8 | 0.920 | 0.958 | 0.849 | 0.010 | 0.900 | 0.937 | 0.872 | 0.033 | 0.793 | 0.825 | 0.613 | 0.034 | 0.934 | 0.973 | 0.920 | 0.015 | 0.844 | 0.888 | 0.749 | 0.042 |
| NIPQ | W8A8 | 0.916 | 0.956 | 0.852 | 0.011 | 0.904 | 0.936 | 0.874 | 0.033 | 0.797 | 0.819 | 0.627 | 0.040 | 0.925 | 0.959 | 0.893 | 0.016 | 0.836 | 0.876 | 0.739 | 0.043 |
| QA-LoRA | W8A8 | 0.921 | 0.936 | 0.843 | 0.017 | 0.930 | 0.953 | 0.909 | 0.022 | 0.865 | 0.895 | 0.732 | 0.015 | 0.912 | 0.938 | 0.862 | 0.020 | 0.847 | 0.872 | 0.739 | 0.035 |
| PEQA | W8A8 | 0.929 | 0.969 | 0.869 | 0.008 | 0.929 | 0.953 | 0.904 | 0.022 | 0.827 | 0.844 | 0.682 | 0.042 | 0.916 | 0.944 | 0.874 | 0.019 | 0.825 | 0.858 | 0.725 | 0.051 |
| QST | W8A8 | 0.918 | 0.939 | 0.835 | 0.014 | 0.899 | 0.925 | 0.862 | 0.036 | 0.799 | 0.819 | 0.616 | 0.039 | 0.881 | 0.909 | 0.811 | 0.030 | 0.811 | 0.845 | 0.684 | 0.049 |
| EQuA (Ours) | W8A8 | 0.925 | 0.961 | 0.856 | 0.009 | 0.919 | 0.948 | 0.890 | 0.026 | 0.834 | 0.848 | 0.680 | 0.029 | 0.919 | 0.945 | 0.876 | 0.020 | 0.838 | 0.869 | 0.734 | 0.040 |
| LSQ | W4A4 | 0.876 | 0.895 | 0.757 | 0.026 | 0.895 | 0.931 | 0.851 | 0.038 | 0.753 | 0.769 | 0.538 | 0.055 | 0.911 | 0.936 | 0.858 | 0.020 | 0.821 | 0.855 | 0.702 | 0.047 |
| NIPQ | W4A4 | 0.884 | 0.908 | 0.781 | 0.023 | 0.901 | 0.934 | 0.863 | 0.035 | 0.771 | 0.791 | 0.561 | 0.043 | 0.919 | 0.952 | 0.878 | 0.019 | 0.826 | 0.872 | 0.711 | 0.046 |
| QA-LoRA | W4A4 | 0.919 | 0.929 | 0.825 | 0.013 | 0.923 | 0.948 | 0.897 | 0.027 | 0.830 | 0.862 | 0.662 | 0.022 | 0.912 | 0.946 | 0.861 | 0.021 | 0.842 | 0.877 | 0.738 | 0.038 |
| PEQA | W4A4 | 0.927 | 0.959 | 0.854 | 0.009 | 0.926 | 0.952 | 0.904 | 0.025 | 0.823 | 0.842 | 0.666 | 0.043 | 0.917 | 0.938 | 0.864 | 0.021 | 0.841 | 0.880 | 0.739 | 0.043 |
| QST | W4A4 | 0.905 | 0.915 | 0.805 | 0.016 | 0.894 | 0.921 | 0.858 | 0.037 | 0.729 | 0.754 | 0.485 | 0.054 | 0.870 | 0.892 | 0.787 | 0.030 | 0.780 | 0.813 | 0.627 | 0.054 |
| EQuA (Ours) | W4A4 | 0.922 | 0.938 | 0.838 | 0.013 | 0.915 | 0.947 | 0.890 | 0.028 | 0.799 | 0.837 | 0.620 | 0.030 | 0.901 | 0.926 | 0.843 | 0.027 | 0.821 | 0.852 | 0.701 | 0.045 |

Table 14: Ablation study on the backbone learning rate of SAM-H.

| Learning Rate | CVC-T | | Kvasir | |
|---|---|---|---|---|
| | $S_\alpha\uparrow$ | $E_\phi\uparrow$ | $S_\alpha\uparrow$ | $E_\phi\uparrow$ |
| $1\times10^{-6}$ | 0.888 | 0.898 | 0.867 | 0.881 |
| $5\times10^{-5}$ | 0.916 | 0.931 | 0.906 | 0.935 |
| $1\times10^{-4}$ | 0.922 | 0.938 | 0.915 | 0.947 |
| $2\times10^{-4}$ | 0.920 | 0.933 | 0.911 | 0.939 |
| $1\times10^{-3}$ | 0.335 | 0.299 | 0.322 | 0.423 |

Table 15: Ablation study on the number of times each block is fine-tuned in each round of BAQF.

| Iteration | CVC-T | | Kvasir | |
|---|---|---|---|---|
| | $S_\alpha\uparrow$ | $E_\phi\uparrow$ | $S_\alpha\uparrow$ | $E_\phi\uparrow$ |
| 10 | 0.915 | 0.930 | 0.915 | 0.940 |
| 50 | 0.921 | 0.939 | 0.916 | 0.945 |
| 100 | 0.922 | 0.938 | 0.915 | 0.947 |
| 200 | 0.922 | 0.946 | 0.916 | 0.944 |

**Polyp Segmentation.** In this task, we include five commonly used datasets: CVC-T (Vázquez et al., 2016), Kvasir (Jha et al., 2020), ETIS (Silva et al., 2014), CVC-612 (Bernal et al., 2015), and CVC-ColonDB (Tajbakhsh et al., 2016). Kvasir includes 1000 images. CVC-612, also called CVC-ClinicDB, includes 612 images. We follow Fan et al. (2020b) to divide the images in CVC-612 and Kvasir into a 9:1 ratio for training and testing. ETIS, CVC-T, and CVC-ColonDB are all used for testing. We adopt $S_\alpha$, $E_\phi$, $F_\beta^\omega$, and MAE as metrics for this task.

**Camouflaged Object Detection.** We choose three commonly used datasets: CAMO (Le et al., 2019), COD10K (Fan et al., 2020a), and NC4K (Lv et al., 2021). CAMO contains 1000 images for training and 250 for testing. COD10K contains 3040 images for training and 2026 for testing. NC4K includes 4121 testing images. We use the training set of CAMO and COD10K for training. We adopt $S_\alpha$, $E_\phi$, $F_\beta^\omega$, and MAE as metrics for this task.

**Leaf Disease Segmentation.** We follow Zhong et al. (2024) to perform the evaluation on this dataset. This dataset includes 498 training samples and 90 testing samples. We adopt mIoU and mDice as metrics for this task.

# I  MORE SEMANTIC SEGMENTATION RESULTS

**Quantitative results.** In Table 12 and Table 13, we report results of SAM-H in the camouflaged object detection task and the polyp segmentation task, which include additional datasets and metrics. The results further verify the effectiveness of our EQuA.

**Qualitative results.** In Fig. 7, Fig. 6, and Fig. 8, We provide visualization results of SAM-H in the three semantic segmentation tasks. The results demonstrate that EQuA achieves competitive or even superior performance in qualitative comparisons.

# J  ADDITIONAL ANALYSIS OF CHANNEL IMPORTANCE ACROSS DIFFERENT DATA DISTRIBUTIONS

We evaluate the stability of our method by computing the proportion of shared important channels across three different data distributions with varying random seeds, as shown in Table 16. The proportion is highest when comparing samples within the same distribution, indicating that weight importance is stable under intra-distribution variations. When comparing across different distributions,

---

**Algorithm 1** EQuA pipeline

---

**Require:** Downstream dataset $\mathcal{D}$, Pseudo-quantized model $\mathcal{F}$
   **for** $i = 1, ..., T$ **do**
      $\mathcal{F}^s \leftarrow split(\mathcal{F})$                                      $\triangleright$ two branches of $\mathcal{F}^s$: $\mathcal{F}_{mib}$ and $\mathcal{F}_{ssa}$
      initialize $B_r$ and $A_r$ for $\mathcal{F}_{ssa}$                               $\triangleright$ if LoRA not initialized
      $\mathcal{L}_1 \leftarrow \mathbb{E}_{x,t \sim \mathcal{D}}[\mathcal{L}(\mathcal{F}^s(x), t)]$
      $B_r \leftarrow B_r - \eta \nabla_{B_r} \mathcal{L}_1$                                    $\triangleright \eta$ : learning rate
      $A_r \leftarrow A_r - \eta \nabla_{A_r} \mathcal{L}_1$
      $s_{w_{ssa}} \leftarrow s_{w_{ssa}} - \eta \nabla_{s_{w_{ssa}}} \mathcal{L}_1$
      $W_{ssa} \leftarrow W_{ssa} + \alpha \cdot A_r B_r$
      $\mathcal{F} \leftarrow merge(\mathcal{F}^s)$
      **if** $i \bmod K = 0$ **then**
         $\mathcal{F}^{fp} \leftarrow disable\_quant(\mathcal{F})$
         **for** $l = 1, ..., L$ **do**
            $\mathcal{L}_2 \leftarrow \mathbb{E}_{x \sim \mathcal{D}}[\|\mathcal{F}_l^{fp}(Y_{l-1}) - \mathcal{F}_l(\hat{Y}_{l-1})\|_2]$
            $s_a \leftarrow s_a - \eta \cdot \nabla_{s_a} \mathcal{L}_2$
         **end for**
      **end if**
   **end for**

---

Table 16: Proportion of shared channels in Top-100 important channels. The results are tested on SAM-H across three different downstream tasks distributions using different random seeds.

| Downstream Tasks | Medical (seed=9582) | Natural (seed=5492) | Agricultural (seed=3639) |
|---|---|---|---|
| Medical (seed=4537) | 0.94 | 0.71 | 0.74 |
| Natural (seed=4234) | 0.72 | 0.90 | 0.77 |
| Agricultural (seed=3638) | 0.76 | 0.70 | 0.91 |

Table 17: Further exploration on channel correlations.

| Strategy | CVC-T $S_\alpha/E_\phi$ | Kvasir $S_\alpha/E_\phi$ |
|---|---|---|
| $A^*$ ($\lambda = 0.01$) | 0.925/0.944 | 0.920/0.946 |
| $A^*$ ($\lambda = 1$) | 0.916/0.934 | 0.910/0.936 |
| A | 0.922/0.938 | 0.915/0.947 |

the proportion decreases, yet a substantial overlap remains. This demonstrates that, although the set of important channels adapts to each distribution, a consistent subset remains important across distributions.

## K   Further Exploration and Future Work

While our EQuA achieves an elegant trade-off between training memory and performance, we will explore further solutions for performance improvement in future work. For example, we attempt to introduce channel correlation by computing the Hessian matrix in Eq. 18 using a Hessian proxy: $X^T X + \lambda I$, and denote this as Strategy $A^*$. The result shown in Table 17 suggests that incorporating channel correlation may have the potential for further improvement, although it may depend on more careful and complex design or hyperparameter selection. On the other hand, our BAQF strategy updates the quantization parameters in the MIB branch by minimizing the MSE loss in a block-wise manner, which does not incorporate information from the task loss. Introducing task-loss gradients into this process may also lead to more effective quantization parameter updates and potentially further improve performance.

## L   The Use of Large Language Models

We use the Qwen3 to help polish sentences and check for spelling errors.

## Ethics Statement

We confirm that this work complies with the ICLR Code of Ethics and does not involve human subjects, sensitive data, or any practices that raise ethical concerns.

## Reproducibility Statement

We describe the components of our method in Sec. 4 and the experimental settings in Sec. 5, and present additional results for verification in Appendix I. We will release the training and testing code, along with the processed datasets, upon publication to ensure full reproducibility.

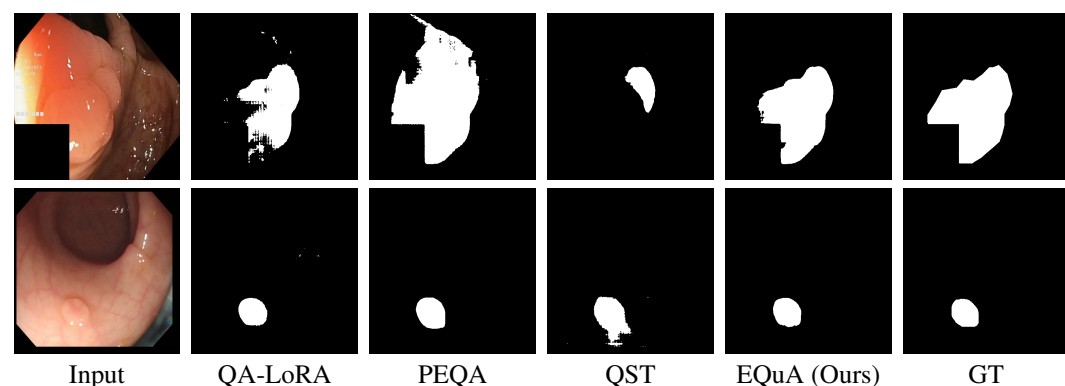

| Input | QA-LoRA | PEQA | QST | EQuA (Ours) | GT |

Figure 6: Visualization results of polyp segmentation of SAM-H under the W4A4 setting. The two cases in the first and second rows are from Kavsir and CVC-T datasets, respectively.

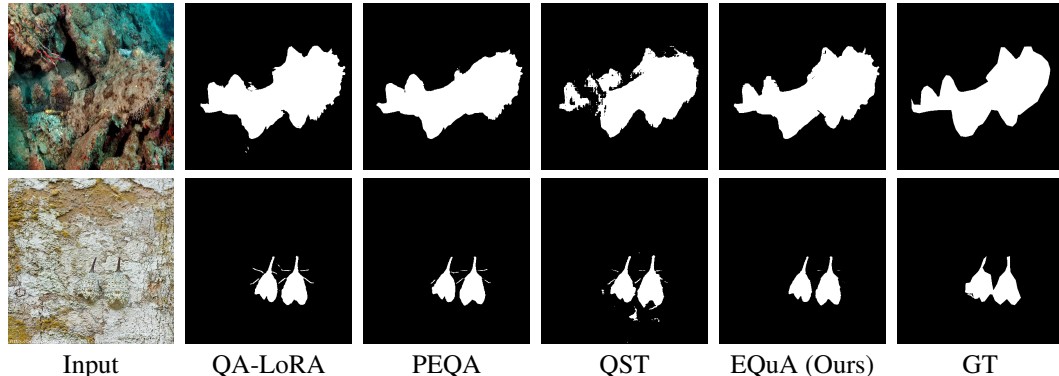

| Input | QA-LoRA | PEQA | QST | EQuA (Ours) | GT |

Figure 7: Visualization results of camouflaged object detection of SAM-H under the W4A4 setting. These cases in the first and second rows are from the CAMO dataset.

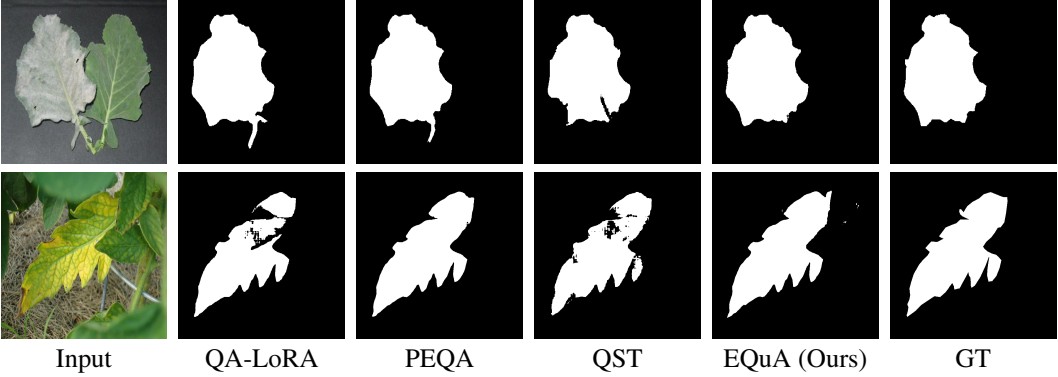

| Input | QA-LoRA | PEQA | QST | EQuA (Ours) | GT |

Figure 8: Visualization results of leaf disease segmentation of SAM-H under the W4A4 setting. These cases in the first and second rows are from the Leaf dataset.

