# OpenReview forum: "Efficient Quantization-Aware Adaptation for Visual Foundation Models"
_ICLR.cc/2026/Conference — Submitted to ICLR 2026_

### Official Review · Reviewer_NEZo · 2025-10-30

**Soundness:** 3
**Presentation:** 3
**Contribution:** 3
**Rating:** 6
**Confidence:** 2

**Summary:**

This paper proposes EQuA (Efficient Quantization-aware Adaptation) for visual foundation models. It addresses high GPU memory use in adaptation and extra deployment computation of existing methods by splitting a lightweight SSA from the backbone (freezing the memory-intensive MIB) and using SAQF/BAQF strategies. EQuA cuts over 70% adaptation memory, keeps competitive performance, and has 1.38× inference speedup. Its contributions include being the first for visual models, the mergeable SSA, SAQF/BAQF, and superior performance-efficiency trade-off via experiments.

**Strengths:**

1. The paper pioneers a mergeable Sub-network Side Adapter (SSA) and SAQF/BAQF strategies for visual foundation models, solving the activation-dominated memory issue unaddressed by LLM-focused methods.
2. Extensive experiments on SAM/ViT across diverse tasks (e.g., segmentation, classification) with strict baselines show EQuA’s 70%+ memory reduction, competitive performance, and 1.38× speedup, ensuring high quality.
3. It clearly explains visual model pain points, details EQuA’s "split-adapt-merge" pipeline, and enables resource-constrained deployment, advancing real-world use of visual models.

**Weaknesses:**

1. The paper only evaluates EQuA on SAM and ViT, lacking tests on other representative visual foundation models like CLIP or Stable Diffusion, limiting the generalizability of its efficiency claims to broader visual task scenarios.
2. While analyzing hyperparameters like ($D_s$) and LoRA rank, the paper provides no discussion on EQuA’s performance under varying input image resolutions or batch sizes beyond 2 and 32, leaving gaps in understanding its efficiency across practical deployment conditions.
3. The paper compares EQuA with QST but does not fully address why its stochastic SSA channel selection is more robust than deterministic top-$D_s$ selection in long-term fine-tuning, with no analysis on overfitting trends across more epochs to validate generalization.

**Questions:**

1. Could you provide supplementary experiments or analysis to explain if EQuA can adapt to visual foundation models like CLIP or Stable Diffusion, and what modifications are needed if not?
2. Could you show how EQuA’s memory, training time, and performance change with different batch sizes and image resolutions, and if these affect hyperparameter settings?
3. Could you supplement ablation studies on overfitting trends of Strategy A vs. Strategy C over more epochs or smaller datasets, and explain why stochastic selection avoids overfitting better?
4. Could you report GC’s additional memory overhead, impact on training throughput, and effectiveness across different models or bit-widths?

---

> ### Author Response · Authors · 2025-11-21
> **Response to Reviewer NEZo**
>
> Thank you for your time with our work. All your concerns are carefully addressed as follows, where Weaknesses are denoted as **W** and Questions are denoted as **Q**. We sincerely hope the revised paper by taken into these valuable comments would meet the expectations of the reviewer, and we are glad to provide further responses if needed.
>
>
> ### **W1&Q1: Adapting EQuA to more visual foundation models**
> We agree that testing on more models would further strengthen our work. Since our method is based on ViT structure, it is in principle applicable to other ViT-based visual foundation models, such as CLIP or DiT. Due to time constraints, however, we hope to leave it as a future work. On the other hand, we believe that SAM and ViT-B are sufficiently representative, which are widely used in prior research \[1*,2*\] to evaluate the generalizability of vision models across various downstream scenarios.
>
>
> ### **W2&Q2: Analysis on different batch sizes and image resolutions**
>
> 1. Different batch sizes: we provide related results of SAM-H and ViT-B across different batch sizes as follows. We have supplemented additional efficiency analysis across different batch size in Table 5 of our revised version.
>
> | SAM-H       | Mem (GB) | Time (hour/epoch) | Kvasir $S\_{\alpha}/E\_{\phi}$ |
> | ----------- | -------- | ----------------- | ---------------------------- |
> | batchsize=2 | 18.5     | 0.52              | 0.915/0.947                  |
> | batchsize=4 | 33.9     | 0.50              | 0.912/0.948                  |
> | batchsize=8 | 65.0     | 0.47              | 0.914/0.941                  |
>
>
> | ViT-B        | Mem (GB) | Time (min/epoch) | Top1 Acc on Cifar100 |
> | ------------ | -------- | ---------------- | -------------------- |
> | batchsize=16 | 1.1      | 0.35             | 62.9                 |
> | batchsize=32 | 1.7      | 0.32             | 63.6                 |
> | batchsize=64 | 3.0      | 0.28             | 63.8                 |
>
>
> 2. Different image resolutions: The pre-trained foundation models were pre-trained with a fixed resolution, and official pre-trained weights for other resolutions are not available. We believe the chosen resolution is representative, since these fixed-resolution pre-trained models are commonly used in the majority of prior studies \[1*, 2*\].
> 3. Affecting hyperparameter settings: We believe that changes in batch size and resolution have only a limited impact on other hyperparameters, since these factors do not play a major role in the effectiveness of the adaptation method.
>
>
> ### **W3&Q3: Ablation studies on overfitting trends**
> We chose Strategy A over Strategy C primarily based on empirical observations. An intuitive explanation is as follows. L2 regularization is often used to constrain large weights, which usually dominate performance but tend to cause overfitting. Channels with higher importance behaves in a similar way and can strongly influence the model output, but they also carry a higher risk of overfitting. Therefore, stochastically sampling channels according to their importance would help reduce the risk, compared with always selecting the top channels. Here, we also report the additional ablation on overfitting trends. When fine-tuned on a sampled Polyp Segmentation dataset, Strategy C shows clearer overfitting trends than Strategy A in test losses. We have supplemented these results in Figure 4 of our revised version.
>
>
> | Epoch                            | 1         | 7         | 13        | 19        | 25        | 30        |
> | -------------------------------- | --------- | --------- | --------- | --------- | --------- | --------- |
> | Strategy A (Train loss/Val loss) | 1.45/0.80 | 0.62/0.73 | 0.45/0.71 | 0.37/0.71 | 0.31/0.70 | 0.29/0.70 |
> | Strategy C (Train loss/Val loss) | 1.38/0.78 | 0.58/0.73 | 0.43/0.69 | 0.35/0.70 | 0.30/0.71 | 0.29/0.73 |
>
>
>
> ### **Q4: Additional ablation of GC across different bit-widths**
> In Table 3 of our paper, we report the additional memory cost introduced by GC. The GC process incurs negligible extra memory overhead. The throughput is reported as follows:
>
> |        | throughput (item/sec) | Mem (GB) |
> | ------ | --------------------- | -------- |
> | SSA    | 0.63                  | 18.6     |
> | SSA+GC | 0.40                  | 18.6     |
>
> We further report the additional ablation of GC on SAM-H across different bit-widths as follows.
>
> |        | bit  | Kvasir $S\_{\alpha}/E\_{\phi}$ |
> | ------ | ---- | ---------------------------- |
> | SSA    | W8A8 | 0.894/0.920                  |
> | SSA+GC | W8A8 | 0.919/0.951                  |
> | SSA    | W6A6 | 0.890/0.915                  |
> | SSA+GC | W6A6 | 0.918/0.945                  |
>
> ### **Reference**
> \[1*\] Zhong Z et al., Convolution meets lora: Parameter efficient finetuning for segment anything model, ICLR 2024.
>
> \[2*\] Mercea O B et al., Time-memory-and parameter-efficient visual adaptation, CVPR 2024.

---

> > ### Comment · Reviewer_NEZo · 2025-11-26
> >
> > Thank you for the response. I don't know much about this field, but I will keep my score.

---

> > > ### Author Response · Authors · 2025-11-26
> > >
> > > We are glad that our responses adequately addressed your concerns.
> > > Thank you for the positive recommendation.

---

### Official Review · Reviewer_wo2P · 2025-10-31

**Soundness:** 2
**Presentation:** 3
**Contribution:** 2
**Rating:** 4
**Confidence:** 4

**Summary:**

EQuA splits ViT blocks into a frozen backbone and a lightweight side adapter, fine-tuning only the adapter with quantization-aware strategies. Alternating SSA fine-tuning and block-wise activation quantizer tuning cuts training memory (>70%) while preserving accuracy, and merges adapters at deployment to avoid extra computation, enabling efficient 4-bit inference.

**Strengths:**

1. This paper directly addresses activation-dominated memory in vision QAT via SSA/MIB splitting and SSA-only backprop, achieving large training memory savings without deployment overhead.
2. The authors provide a sound, reproducible design: activation-scaled channel selection, gradient checkpointing to inject backbone signals, and LoRA-in-SSA for parameter efficiency.
3. Experiments are comprehensive and convincing: competitive accuracy with >70% memory reduction, outperforming QST, and preserving 4-bit speedups validated by CUTLASS-based latency.

**Weaknesses:**

1. The channel selection strategy lacks detailed ablation. It lacks performance–memory trade-off curves across channel split ratios p and comparisons of alternative importance proxies (e.g., |W|, E[|A|], E[A^2]|W|, gradient-based metrics).
2. SAQF partitions the weight by channel without theoretical motivation or justification. I wonder how the authors consider its potential harm for inter-channel correlations. Furthermore, the authors do not assess whether the channel importance remains stable across different data distributions or input domain.
3. The method relies on alternating SAQF and BAQF, which raises concerns about training stability and may require careful manual tuning of the alternation frequency, number of steps, and learning-rate schedule. Could the authors provide an analysis of the robustness of this strategy?

**Questions:**

Please see Weaknesses.

---

> ### Author Response · Authors · 2025-11-21
> **Response to Reviewer wo2P**
>
> Thank you for your time with our work. All your concerns are carefully addressed as follows, where Weaknesses are denoted as **W**. We sincerely hope the revised paper by taken into these valuable comments would meet the expectations of the reviewer, and we are glad to provide further responses if needed.
>
> ### **W1: Detailed ablation**
> 1. Trade-off curves across channel split ratios p: Thank you for your suggestion. We have added an ablation study about the channel split ratios p curves on SAM-H (W4A4) in Figure 4c of our revised version.
>
> | channel split ratios p ($\frac{D\_s}{D}$) | Mem (GB) | Kvasir $S\_{\alpha}$ |
> | ---------------------------------------- | -------- | ------------------- |
> | 0.03                                     | 18.3     | 0.906               |
> | 0.1                                      | 18.8     | 0.917               |
> | 0.3                                      | 20.1     | 0.917               |
> | 0.5                                      | 21.0     | 0.919               |
> | 0.7                                      | 23.2     | 0.920               |
> | 1.0                                      | 25.3     | 0.919               |
>
>
> 2. Alternative importance proxies: we did provide discussion on six alternative importance proxies in Appendix B in our original version, which we believe are sufficiently representative. As suggested, we further report an ablation on these additional proxies. We have supplemented more results in Table 6 of our revised version.
>
> | Strategy          | Mem (GB) | Kvasir $S\_{\alpha}$ |
> | ----------------- | -------- | ------------------- |
> | Ours              | 9.6      | 0.915               |
> | $\|W\|$           | 0.0      | 0.908               |
> | $E[\|A\|]$        | 9.6      | 0.906               |
> | $E[\|A\|^2]\|W\|$ | 9.6      | 0.912               |
> | gradient-based    | 57.2     | 0.917               |
>
>
> ### **W2-1: Theoretical analysis and inter-channel correlations**
> We did provide the mathematical details in Appendix A in our original version, which provide a theoretical motivation. We agree that fully modeling inter-channel correlation could further improve performance. Nevertheless, capturing such correlations would result in substantial computational or memory overhead, such as the computation of the Hessian matrix. To maintain efficiency, our method is based on a reasonable channel independence assumption to simplify the complexity of the problem, which has also been adopted widely in prior works \[1*\].
>
>
> ### **W2-2: The stability of channel importance across different data distributions**
> We assess the stability by computing the proportion of shared important channels in Top-100 important channels across three different data distributions with different random seeds.
> The proportion is highest within the same distribution, indicating stable behavior under intra-distribution variations. Although the proportion decreases across distributions, a substantial overlap remains (i.e., $>70\\%$). This suggests that while the set of important channels adapts to each distribution, a consistent subset is stable and remains important across distributions. We have supplemented this analysis in Appendix J and Table 16 of our revised version.
>
> |              | Medical | Natural | Agricultural |
> | ------------ | ------- | ------- | ------------ |
> | Medical      | 0.94    | 0.71    | 0.74         |
> | Natural      | 0.72    | 0.90    | 0.77         |
> | Agricultural | 0.76    | 0.70    | 0.91         |
>
>
>
> ### **W3: Ablation on hyper-parameters: Frequency, Number of steps, and Learning-rate**
> 1. Alternation frequency $K$: we did provide ablation study in Table 7 in our original version, which demonstrate the training stability when $K$ is set between 5 and 15.
> 2. Number of steps (*Iteration* in Table 11 of our revised version): we provide an additional ablation (SAM-H W4A4) on the Iteration. Our BAQF strategy is robust when Iteration is set between 50 and 200. We have supplemented more results in Table 15 of our revised version.
>
> | Iteration | Kvasir $S\_{\alpha}$ |
> | --------- | ------------------- |
> | 10        | 0.915               |
> | 50        | 0.916               |
> | 100       | 0.915               |
> | 200       | 0.916               |
>
>
> 3. Learning-rate: We report an ablation about the learning rate schedule. Our learning rate choice is generally robust when it is within the range of $5\times10^{-5}$ to $2\times10^{-4}$. We have supplemented more results in Table 14 of our revised version.
>
> | learning rate    | Kvasir $S\_{\alpha}$ |
> | ---------------- | ------------------- |
> | $1\times10^{-6}$ | 0.867               |
> | $5\times10^{-5}$ | 0.906               |
> | $1\times10^{-4}$ | 0.915               |
> | $2\times10^{-4}$ | 0.911               |
> | $1\times10^{-3}$ | 0.322               |
>
> ### **Reference**
> \[1*\] Mingjie Sun et al., A simple and effective pruning approach for large language models, ICLR 2024.

---

> > ### Comment · Reviewer_wo2P · 2025-11-27
> >
> > Thank you for your rebuttal, but it hasn't fully addressed my concerns. Regarding the theoretical analysis and inter-channel correlations, the authors suggest that modeling channel correlations could lead to further performance improvements. It would be valuable to see comparative experiments demonstrating this effect. Have the authors explored this direction?

---

> ### Author Response · Authors · 2025-11-28
>
> Thank you for your response.
>
> Following the reviewer’s suggestion, we make a further exploration on channel correlations. We preliminarily attempt to introduce channel correlation by computing the Hessian matrix $H$ in Eq. 18 of our paper using a Hessian proxy [1\*]: $X^TX$ (the Gram matrix of the input activation $X$), which is a simplified approximation of the Hessian matrix. Therefore, the weight importance $M\_{i}$ can be calculated as follows:
>
> $Strategy\\, A^*: M\_{i}=\sum\_{j=1}^{D\_{out}}\frac{W\_{ij}^2}{(H^{-1})\_{ii}}=\sum\_{j=1}^{D\_{out}}\frac{W\_{ij}^2}{((X^TX)^{-1})\_{ii}}\approx\sum\_{j=1}^{D\_{out}}\frac{W\_{ij}^2}{((X^TX+\lambda I)^{-1})\_{ii}}$
>
> where $\lambda $ is an extra hyperparameter that ensures numerical stability by making the matrix well-conditioned and always invertible. Then we use this weight importance (Strategy $A^*$) in our EQuA and report the results as follows.
>
>
>
>
> |                                           | Time     | Memory  | CVC-T $S_{\alpha}/E_{\phi}$ | Kvasir $S_{\alpha}/E_{\phi}$ |
> | ----------------------------------------- | -------- | ------- | --------------------------- | ---------------------------- |
> | Strategy $A^*$ ($\lambda=1\times10^{-2}$) | 55.2 sec | 12.9 GB | **0.925/0.944**             | **0.920**/0.946              |
> | Strategy $A^*$ ($\lambda=1$)              | 55.2 sec | 12.9 GB | 0.916/0.934                 | 0.910/0.936                  |
> | Ours                                      | 20.6 sec | 9.6 GB  | 0.922/0.938                 | 0.915/**0.947**              |
>
>
>
> This result suggests that designs that incorporate channel correlation have the potential for further improvement, and we believe that more significant gains could be achieved under more careful design and hyperparameter selection, which may require more time than the rebuttal though.
>
> In summary, we thank the reviewer again for pointing out this promising direction for future work. Meanwhile, we sincerely hope that the final recommendation can be based on the established contributions of our current work, which has already been well recognized.
>
> [1*] Nagel M et al., Up or Down? Adaptive Rounding for Post-Training Quantization, ICML 2020.

---

### Official Review · Reviewer_mEHg · 2025-11-01

**Soundness:** 3
**Presentation:** 3
**Contribution:** 3
**Rating:** 8
**Confidence:** 3

**Summary:**

The authors propose EQuA, which splits the network into a Memory-Intensive Backbone (MIB) and a lightweight Sub-network Side Adapter (SSA), blocks backprop through the MIB to remove large activation checkpoints, and alternates two procedures: SAQF (side-adapter QAT) and BAQF (block-wise activation-quantizer finetuning). At deployment, the SSA is merged back into the backbone, incurring no extra inference compute. Experiments report large training-memory reductions (>70% in some cases) with competitive accuracy, and up to 1.38× inference speedups at low bit-widths.

**Strengths:**

1. This work brings the reparameterization idea to adaptation by blocking gradients through the memory-intensive backbone and routing them through a lightweight side adapter (alternating SAQF/BAQF). It directly tackles the true QAT memory bottleneck while staying deployment-aligned, since the adapter is merged back into the backbone at inference.
2. The approach shows large, consistent gains: over 70% reduction in training memory with accuracy comparable to strong baselines, and up to ~1.38× end-to-end inference speedups at low bit-widths across multiple models and tasks.

**Weaknesses:**

1. Error modeling is very interesting. However, in this paper the authors did not further analyze the relation between sub-net design and error propagation. This makes it hard to understand why the authors choose the proposed sub-net structure.
2. The task distribution might influence the weight split results. How do we understand the pros and cons of the proposed methods when we need to adapt to a batched tasks?

**Questions:**

Please see the weaknesses.

---

> ### Author Response · Authors · 2025-11-21
> **Response to Reviewer mEHg**
>
> Thank you for your time with our work. All your concerns are carefully addressed as follows, where Weaknesses are denoted as **W**. We sincerely hope the revised paper by taken into these valuable comments would meet the expectations of the reviewer, and we are glad to provide further responses if needed.
>
> ### **W1: Relationship between SSA design and error propagation**
> We made some related discussions on the side adapter and error propagation in Appendix D in our original version. The primary purpose of our sub-network side adapter (SSA) design is to reduce memory overhead. Based on the discussion in Appendix D, we argued that this side adapter structure may be affected by error propagation. Therefore, we design other components such as BAQF and gradient checkpoints to mitigate this effect.
>
> ### **W2: Pros and Cons when handling batched task distributions**
> We assess the weight-splitting results by computing the proportion of shared important channels across three different task distributions using different random seeds, as shown in the following table.
> When comparing different task distributions, the overlap of the weight-splitting results decreases compared to that within the same task distribution, yet a considerable overlap still remains. This suggests that our method can both identify channels with cross-task generalizability and distinguish task-specific differences, which may be beneficial for batched tasks. Nevertheless, our method is not explicitly designed for batched scenarios, and appropriately handling task-specific differences in batched tasks remains an open challenge. We leave it as a future work.
>
> | $\frac{\text{Number of shared channels in Top-100 important channels}}{100}$ | Medical (seed=9582) | Natural (seed=5492) | Agricultural (seed=3639) |
> | ---------------------------------------------------------------------------- | ------------------- | ------------------- | ------------------------ |
> | Medical (seed=4537)                                                          | 0.94                | 0.71                | 0.74                     |
> | Natural (seed=4234)                                                          | 0.72                | 0.90                | 0.77                     |
> | Agricultural (seed=3638)                                                     | 0.76                | 0.70                | 0.91                     |

---

### Official Review · Reviewer_qotj · 2025-11-10

**Soundness:** 3
**Presentation:** 2
**Contribution:** 2
**Rating:** 4
**Confidence:** 5

**Summary:**

This paper proposes a framework (EQuA) for joint quantization and adaptation tailored for visual foundation models. The framework splits model weights based on importance, obtaining a sub-network component (adapter) called SSA for gradient propagation and updates, while other weights only undergo forward propagation. This approach reduces intermediate activations required for gradient computation. To enhance model training, two strategies are introduced: SAQF for optimizing SSA and BAQF for fine-tuning the merged weights, thereby minimizing quantization errors. Experimental results demonstrate that this method outperforms baseline QST, while reducing memory usage during training compared to other SOTA methods.

**Strengths:**

1. The selection of a subset of weights (SSA) for quantized fine-tuning is sound and novel (Eq.6 & Eq.7).
2. EQuA reduces memory usage during training.
3. In the visualizations presented in the paper, EQuA yields ​​favorable​​ results.
4. The strategy of splitting and merging for SSA and MIB is flexible and can be used for further exploration in other researches.

**Weaknesses:**

1. The paper lacks an overall description in the method section, particularly for the ​​initial quantization​​. For instance, the training pipeline is unclear. It omits how the initial model is prepared (e.g., PTQ, QAT, or a full-precision model) before applying EQuA fine-tuning. The paper should also specify the source for quantizer: are the quantization parameters for the MIB and SSA weights calibrated independently on their respective subsets, or jointly on the original, full-precision weight tensor? Additionally, Figure 2 fails to illustrate the quantizer, which hinders readability.
2. It is important to report the cost of the weight importance calculation and SSA selection, detailing the computational cost (e.g., time) and resource footprint (e.g., memory) incurred by this stage.
3. The practical advantage of using LoRA is questionable. As shown in Table 3, the significant reduction in trainable parameters does not result in savings in training time or memory, while slightly hurting performance.
4. The robustness of the SSA selection strategy requires further demonstration. Since the method relies on probabilistic sampling of SSA channels, the results of EQuA may be unstable. Reporting the standard deviation over multiple experimental runs would provide more compelling evidence (a small table is enough).

**Questions:**

The paper presents an interesting approach for efficient fine-tuning of foundation models. However, **the practical advantages and core motivation of the EQuA framework** require *further clarification*, particularly from a deployment-centric perspective. For real-world applications of foundation models, the primary concerns are typically inference efficiency and maintained model performance, rather than time / memory in training. While EQuA reduces training memory, it does not lead to a significant improvement in training time due to the overhead of managing partially fine-tuned weights. Moreover, memory constraints can often be alleviated through practical means like training with more GPUs​, hardware upgrades or gradient accumulation without compromising the final model capability.

Some limitations of EQuA related to my question:
- According to Table 5, EQuA offers no inference advantage over PEQA, with similar training time.
- More critically, as shown in Table 1, the method incurs significant performance degradation on certain datasets (e.g., >2% drop on CAMO) compared with PEQA, raising concerns about its practicality.
- Furthermore, the trade-off explored in Table 4 appears problematic: adjusting $D_{s}$ to improve training efficiency comes at the cost of a substantial performance drop, making it difficult to identify a viable solution.

---

> ### Author Response · Authors · 2025-11-21
> **Response to Reviewer qotj (Part 1/2)**
>
> Thank you for your time with our work. All your concerns are carefully addressed as follows, where Weaknesses are denoted as **W** and Questions are denoted as **Q**. We sincerely hope the revised paper by taken into these valuable comments would meet the expectations of the reviewer, and we are glad to provide further responses if needed.
>
> ### **W1: Clarity of description**
> Thank you for your suggestion. We have provided a clearer description as suggested in our revised version.
> 1. The initial preparation: We did explain the preparation of the initial model in Line313 in our original version. We use RepQ-ViT to quantize models as a PTQ initialization before applying EQuA fine-tuning.
> 2. The source for quantizer: We perform the PTQ  initialization by jointly calibrating quantization parameters on the original full-precision weights. We have provided a clearer description in Line313-315 of our revised version.
> 3. The illustration of quantizers in Figure 2: We have introduced the quantizers in Figure 2 in our revised version.
>
> ### **W2: The cost of the weight importance calculation and SSA selection**
> Thank you for your suggestion. Since this process is achieved solely through a one-step forward before EQuA fine-tuning, and does not require backward gradients, its time cost is negligible, and its memory cost is significant lower than that of training. We have added these costs in Table 7 of our revised version, and we also provide them as follows:
>
> | batch size   | time (sec) | Memory (GB) |
> | ------------ | ---------- | ----------- |
> | 2            | 20.6       | 9.6         |
> | 4            | 33.2       | 16.4        |
> | 8            | 57.6       | 30.2        |
>
>
> ### **W3: The advantage of using LoRA**
> The significant reduction in trainable parameters using LoRA demonstrates practical advantages during downstream task switching at deployment. When we need to save multiple downstream task variants for a pre-trained model, we can just save fine-tuned parameters trained on different downstream tasks. By using LoRA, we further reduce the number of fine-tuned parameters that needed to be saved by 75% in Table 3. At deployment phase, when switching downstream tasks, we can just switch the LoRA parameters to achieve efficient model switching for different downstream tasks. We have clarified the advantage in Line448-449 of our revised version.
>
>
> ### **W4: The robustness of the SSA selection strategy**
> Thank you for your suggestion. We have supplemented discussion about three experimental runs by using different random seed in Table 8 of our revised version. we report the mean and deviation results as follows, which can verify the robustness.
>
> |      | CVC-T $S\_{\alpha}/E\_{\phi}$       | Kvasir $S\_{\alpha}/E\_{\phi}$      | ETIS $S\_{\alpha}/E\_{\phi}$        |
> | ---- | --------------------------------- | --------------------------------- | --------------------------------- |
> | mean$\pm$std | $0.924\pm0.005$ / $0.940\pm0.004$ | $0.917\pm0.002$ / $0.945\pm0.002$ | $0.795\pm0.003$ / $0.833\pm0.003$ |

---

> ### Author Response · Authors · 2025-11-21
> **Response to Reviewer qotj (Part 2/2)**
>
> ### **Q1: The practical advantages and core motivation**
> We agree that "inference efficiency and model performance" are primary concerns for real-world applications of foundation models, but we respectfully cannot agree that the training memory is LESS important. In fact, **the training memory of foundation models remains a significant barrier for researchers and organizations with limited computing budgets**. For example, an 80GB A100 GPU is much more expensive than a 24GB RTX 3090 (e.g., \\$15000 of A100 vs. \\$1600 of RTX3090), and also much more costly on cloud or rental platforms (e.g., \\$0.69 per hour of A100 vs. \\$0.14 per hour of RTX3090). Therefore, the reduction of training memory could significantly reduce the cost of training foundation models, and has attracted plenty of research attentions recently\[1*,2*\].
>
> Our work provides a novel perspective for achieving an elegant trade-off by reducing the training memory of foundation models, enabling them to be trained and deployed end-to-end at lower cost. We believe this contributes to making the development and real-world application of foundation models more accessible to a broader community.
>
>
> ### **Q2: Advantage over PEQA**
> As discussed in Q1 regarding the importance of training cost, our EQuA has advantage over PEQA in training efficiency. Although PEQA achieves SOTA performance, its training efficiency is fundamentally limited by the characteristics of visual foundation models, where activations rather than floating-point weights result in large memory consumption. Our EQuA offers a substantial reduction in training cost, achieving **over 73% memory savings** along with a **slight reduction in training time**. Moreover, since both PEQA and EQuA ultimately produce quantized models with the same efficient inference structure, we share the same inference advantage. In comparison, another related memory-efficient baseline, QST, sacrifices both inference efficiency ($1.38\times$ of our EQuA vs. $1.18\times$ of QST) and performance (e.g., $>7\\%$ drop on CAMO compared to PEQA).
>
> ### **Q3: Hyper-parameters $D_s$ adjustment**
> The performance only drops noticeably when the hyper-parameter $D_s$​ is too small (e.g., $D_s$=32 in Table 4). As $D_s$​ increases further, the performance improvement exhibits diminishing returns. In Figure 4c of the revised version, we include a performance-memory curve with respect to the channel-split ratio, which shows that a split ratio of around 10% (i.e., $D_s$ between 64 and 128) can provide a practical solution for performance-memory trade-off.
>
> ### **Reference**
> \[1*\] Dettmers T et al., Qlora: Efficient finetuning of quantized llms. NeurIPS 2023.
>
> \[2*\] Zhang Z et al., Quantized side tuning: Fast and memory-efficient tuning of quantized large language models, ACL 2024.

---

> ### Comment · Reviewer_qotj · 2025-11-26
>
> Thank you for the detailed reply and additional experiments.
>
> *Weaknesses*
>
> Regarding W1, my comment aims to improve the **method section**. The method section lacks clarity and can only be fully understood when combined with​ the experiment section. I appreciate the revisions, and they do not affect my overall score. I have no further questions regarding the weaknesses.
>
> *Questions*
>
> I completely agree with the importance of Memory-Efficient Fine-Tuning. Methods like QLoRA, PEQA, and QST have all made significant improvements in this area. While reducing training memory requirements, these methods also decrease training time.
> For EQuA, it seems to only offer a trade-off between training memory and performance. I believe such a compromise might not be suitable for foundation models.
>
> In fact, I'm particularly concerned about the performance degradation, as EQuA trains significantly more parameters than PEQA, yet still shows a noticeable performance gap.
> While **I generally support the approach proposed in the paper**, the observation that EQuA's method (training partial weights + partial quantization parameters) cannot surpass PEQA's approach (only training quantization parameters) indicates that the quantization error of non-critical weights/channels (or inter-channel correlations) is also highly important. There is still room for improvement.
> I hope the authors could provide some analysis on this issue and explore whether there are better solutions.

---

> ### Author Response · Authors · 2025-11-26
>
> Thank you for your response. We are glad to hear that all previous weaknesses have been adequately addressed, and we are encouraged that the reviewer “generally supports the approach proposed in the paper”.
>
> Now there is only one remaining question: while “EQuA seems to only offer a trade-off between training memory and performance, there is still room for improvement”.
>
> Following the suggestion of the reviewer, we would like to provide more analysis on this issue:
>
> (1)	Our method NOT ONLY provides a trade-off between training memory and performance, but also decreases training time compared to PEQA. As shown in Tables 3 and 5, the training time of our method is 0.52 hour/epoch, which is 10% lower than that of PEQA (0.58 hour/epoch). In contrast, the performance drop compared to PEQA is only 2% on CAMO, with a 73% memory saving.
>
> (2)	We agree that our method may still have room for performance improvement. We think a key aspect lies in the way the quantization parameters are updated. The PEQA updates quantization parameters though gradients with respect to task loss, while BAQF strategy of our method updates the quantization parameters in the MIB branch by minimizing the MSE loss of each ViT block, i.e., the loss between the floating-point output and the quantized output, which does not incorporate information from the task loss. Therefore, a possible improvement is to introduce the gradient with respect to the task loss into the update process of quantization parameters in the MIB branch. For example, under the SAQF strategy, we could alternately set a small subset of modules in the MIB branch as learnable, thereby introducing the gradient with respect to the task loss to update the quantization parameters in these modules. We will further look into this in our future work and have added corresponding discussions in Line1106-1113 of our revised paper.
>
> (3)	Last but not least, **we believe the key value of our work lies in exploring a new direction for joint and efficient fine-tuning and deployment of visual foundation models, which itself could be more insightful than simply achieving state-of-the-art performance**.
>
> Once again, we would like to express our great gratitude to the reviewer for helping us clarify the paper and encouraging future directions. We sincerely hope that the final recommendation can be based on the established contributions of our current work, which has already been well recognized.

---

> > ### Comment · Reviewer_qotj · 2025-11-26
> >
> > I appreciate the authors' prompt response. I need some time to review the paper and the authors' reply again. If the authors have any additional clarifications, you are welcome to provide them.

---

### Official Review · Reviewer_XqgE · 2025-11-12

**Soundness:** 2
**Presentation:** 2
**Contribution:** 2
**Rating:** 4
**Confidence:** 5

**Summary:**

The paper proposed a method (EQuA) to reduce memory footprint for fine-tuning vision foundation models, like SAM or ViT. The main challenge in this task is that the memory consumption is dominated by cached activations needed for backward prop as opposed to the size of the model, i.e. number of parameters, which means, typical model compression + tuning scheme, like QA-LoRA, will not be effective in terms of reducing the peak memory usage. The proposed method first identifies important weights and sliced a small portion of $W_V$, $W_{proj}$ in self-attention module and $Lin_1$, $Lin_2$ in MLP module into a side branch (SSA) while the majority will be kept in the a backbone branch (MIB). Although tuning only SSA while keeping MIB frozen could reduce the cached activations, in order to maintain the model performance/quality, a few additional techniques were also employed, i.e. gradient checkpointing, LoRA, and block-wise activation quantization fine-tuning (BAQF). Experimental results of ViT-B on VTAB (Table 2) and SAM-H on medical, natural, and agricultural benchmarks (Table 1) are mostly comprarable to QA-LoRA, with a few specific tasks at ~3% lower than QA-LoRA while peak memory usage is ~1/4 compared to QA-LoRA.

**Strengths:**

1. reasonable amount of experiments and ablation studies.
2. relevant and useful details are provided in Appendix.

**Weaknesses:**

**1. gradient checkpointing**

In Section 4.3, at first, the author emphasized that the reduction in memory usage was mainly because backward gradients were only calculated in the side branch, while backward propagation through the backbone branch was intentionally bypassed. Consequently, the cached activations were on the order of N × B × Ds instead of N × B × D. However, immediately on Line 279 the author stated that the gradients in backbone branch were still needed and computed using gradient checkpointing. In other words, the memory saving and the reduction in cached activations seems to be mainly achieved by gradient checkpointing and not because of the splitting of the side branch. This section together with Table 3 ablation study may cause quite some confusions and may need further clarifications. For example, some discussions to the following questions should be included:

- Assuming "SSA only" condition (row 2) in Table 3 refers to the case where the gradient computation in backbone branch was entirely skipped. This table should also include the condition in which the backbone gradients were computed and gradient checkpointing was not employed. This condition is expected to cache all the activations in both side branch and backbone branch, which should have similar memory consumption with reference methods. This would give readers a comparison with the SSA+GC condition (row 3) in Table 3.

- If gradient checkpointing is applied to other reference methods, e.g. QA-LoRA, in a similar way, what would be the memory consumption?

- Since gradient checkpointing still needs to cache some activations in backbone, if Table 3 row 2 (SSA-only) does not enable the gradients computation in backbone branch, why would Row 3 (SSA+GC) have the same memory consumption as Row 2?


**2. Q K layers**

In the proposed scheme, Q and K in the self-attention modules are not sliced like the other linear layers. It would be helpful for readers if the author added a few sentences in Section 4.2 explaining the reasons or challenges behind keeping Q and K unsliced.


**3. Inference efficiency**

As shown in Table 5, memory footprint at inference stage is much smaller than fine-tuning stage. Therefore, even for a GPU with limited memory can afford a range of batch size choices. Batch size dependency is a very useful information when discussing the latency and inference efficiency. Author may want to consider including a few data points with different batch size for Table 5, or separate the inference part as a line plot with batch size on the x-axis.

**Questions:**

please see Weaknesses above

---

> ### Author Response · Authors · 2025-11-21
> **Response to Reviewer XqgE (Part 1/2)**
>
> Thank you for your time with our work. All your concerns are carefully addressed as follows, where Weaknesses are denoted as **W**. We sincerely hope the revised paper by taken into these valuable comments would meet the expectations of the reviewer, and we are glad to provide further responses if needed.
>
> ### **W1-1: Clarifications about memory saving, gradient checkpointing, and side branches**
>
> There might be some misunderstandings here. We would like to further clarify the differences between our design and the regular gradient checkpointing (GC) operation:
> 1. Regular GC operation (Two Step): Regular GC first performs an additional forward propagation at the backward phase to recompute the intermediate activations. This process serves as the core mechanism for memory savings in regular GC. It then performs a backward propagation that uses these activations to recompute gradients of both the module’s inputs and trainable parameters.
> 2. GC operation in our design (One Step): Our GC only recomputes the gradients of the inputs (denoted as $X^V_{mib}$ and $X_{mib}^{Lin1}$) of the V layer and the Lin1 layer in MIB branch. Since we split the SSA and MIB branches, the MIB branch is excluded from the computation graph (the reason for memory savings) and therefore contains no trainable parameters. **This design enables a one-setp computation for the gradient recomputation** of $X^V_{mib}$ and $X_{mib}^{Lin1}$:
>
>
> $\frac{\partial L}{\partial X^{V}\_{mib}}=[A^T((\frac{\partial L}{\partial Y^{Proj}\_{mib}}W^{Proj,T}\_{mib})\odot \mathbf{1}^{Proj}\_{mib})\odot \mathbf{1}^V\_{mib}]W^{V,T}\_{mib}$
>
>
> $\frac{\partial L}{\partial X^{Lin1}\_{mib}}=(\frac{\partial L}{\partial Y^{Lin2}\_{mib}}W^{Lin2,T}\_{mib}\odot \mathbf{1}^{Lin2}\_{mib})\phi(X^{Lin1}\_{mib}W^{Lin1}\_{mib})W^{Lin1,T}\_{mib}\odot \mathbf{1}^{Lin1}\_{mib}$
>
>
> where $\phi(\cdot)$ is the analytical gradients of GeLU function. $\mathbf{1}^{V}\_{mib}$, $\mathbf{1}^{Proj}\_{mib}$, $\mathbf{1}^{Lin1}\_{mib}$, and $\mathbf{1}^{Lin2}\_{mib}$ are boolean mask tensors from activation quantizers of the related layer. These tensors, except for $\mathbf{1}^{Proj}\_{mib}$ and $\mathbf{1}^{Lin2}\_{mib}$, are all already shared in the SSA branch and does not need to be recomputed.  We manually cache $\mathbf{1}^{Proj}\_{mib}$ and $\mathbf{1}^{Lin2}\_{mib}$ during the forward pass before loss computation, and their memory cost as boolean tensors is negligible, approximately 1% of the total cost.
> In summary, **memory savings are achieved though the split of SSA and MIB, while the GC in our design specifically refers to this one-step computation and merely maintains (rather than causes) the memory savings**.
> We have supplemented this clarification in Appendix E (Line958-1001) of our revised version.
> In addition, we have also revised the statement in Line274-277 in our revised version (Line274-277 in the original version) and do not affect the conclusions of our work.
>
>
> ### **W1-2: Ablation where the backbone gradients were computed without gradient checkpointing**
> We have supplemented this additional ablation item in Table 3 of our revised version, and we also report it as follows. "$SSA^\*$" denotes the suggested condition. The results show that memory usage of "$SSA^\*$" is significant larger than that of “SSA”, indicating that our "SSA" causes the memory savings. Comparing the "SSA" and "SSA+GC" conditions, the memory remains unchanged under the "SSA+GC" condition, indicating that GC maintains (rather than cause) the memory savings.
>
> |         | Param (M) | Train (hour/epoch) | Mem (GB) | Kvasir $S\_{\alpha}$ |
> | ------- | --------- | ------------------ | -------- | ------------------- |
> | Base    | 641.46    | 0.60               | 72.4     | 0.895               |
> | +$SSA^\*$ | 29.90     | 0.57               | 67.6     | 0.918               |
> | +SSA     | 29.90     | 0.32               | 18.6     | 0.881               |
> | +SSA\&GC | 29.90     | 0.50               | 18.6     | 0.910               |
>
>
>
> ### **W1-3: Applying gradient checkpointing to QA-LoRA**
> Applying GC to QA-LoRA can lead to a comparable memory consumption to EQuA. However, the training time becomes significantly longer, adding more than 50\% additional time and thus sacrificing training efficiency, since QA-LoRA with GC has to recomputing additional gradients of all trainable parameters in the ViT modules, which involves recomputing a large number of activations multiplied with these learnable weights.
>
> |                     | Mem (GB) | Train (hour/epoch) | Kvasir $S\_{\alpha}/E\_{\phi}$ |
> | ------------------- | -------- | ------------------ | ---------------------------- |
> | QA-LoRA             | 68.9     | 0.55               | 0.923/0.948                  |
> | QA-LoRA +regular GC | 16.4     | **0.83**           | 0.923/0.951                  |
> | Ours                | 18.5     | 0.52               | 0.915/0.947                  |

---

> ### Author Response · Authors · 2025-11-21
> **Response to Reviewer XqgE (Part 2/2)**
>
> ### **W1-4: Explanation about the same memory consumption of Row2 and Row3 in Table3**
> As discussed in W1-1, tensors required by our GC, except for $\mathbf{1}^{Proj}\_{mib}$ and $\mathbf{1}^{Lin2}\_{mib}$, are shared and used in the SSA branch. In our code implementation, $\mathbf{1}^{Proj}\_{mib}$ and $\mathbf{1}^{Lin2}\_{mib}$ are manually cached (only 0.2 GB, 1\% of the total memory usage) during the forward pass before loss computation, even without enabling GC, which results in negligible difference in memory consumption between SSA and SSA+GC in Table 3.
>
> ### **W2: Explanation about the design of Q and K layers**
> Thank you for your suggestion. We have added an explanation in Line235 of our revised version.
> We only split $W^V$ (the weights of layer V) because the output of MHSA can be decomposed into a block matrix multiplication form in Eq. 6. However, $W^Q$ and $W^K$ (the weights of layers Q and K) are used to compute the attention map, and this computation is non-linear and cannot be directly decomposed into a block matrix form.
>
>
> ### **W3: More cases of different batch size**
> Thank you for your suggestion. We have added more cases about batch size and inference efficiency in Table 5 of our revised version. We list some additinal results (sec/batch) as follows:
>
> | batch size  | 1    | 2    | 4    |
> | ----------- | ---- | ---- | ---- |
> | Full        | 0.75 | 1.44 | 2.94 |
> | SAM-Adapter | 0.88 | 1.67 | 3.38 |
> | PEQA        | 0.54 | 1.06 | 2.14 |
> | QST         | 0.63 | 1.24 | 2.49 |
> | EQuA        | 0.54 | 1.06 | 2.14 |

---

### Author Response · Authors · 2025-12-02
**Concluding Remarks**

We appreciate the efforts of the AC and all reviewers in reviewing our submission and discussion. While the discussion period comes to the end, we would like to briefly summarize the current status.


### Reviewer mEHg (initial rating: 8)

The reviewer gives highly positive comments on the idea (“**directly tackles the true QAT memory bottleneck**”) and the effectiveness (“**large, consistent gains**”) of our method. We have made further explanation regarding the minor issues.


### Reviewer NEZo (initial rating: 6)

The reviewer gives highly positive comments on our work (“**Extensive experiments across diverse tasks**” and “**clearly explains visual model pain points**”). After discussion, the reviewer keeps the positive score.


### Reviewer qotj (initial rating: 4)

After discussion, the reviewer now “**has no further questions regarding the weaknesses**” and “**generally supports the approach proposed in the paper.**” The only remaining open question is “EQuA seems to only offer a trade-off between training memory and performance, and there is still room for improvement.”
However, our method NOT ONLY provides a trade-off between training memory and performance but also decreases training time compared to the main competitor (PEQA) by 10%, as evidenced in the original paper (Table 5). Thus, we do not see this as a shortcoming.
In addition, following the reviewer’s suggestion, we analyze the potential room for improvement as future work in our revised paper (Appendix K).
Last but not least, we believe the key value of our work lies in **exploring a new direction** for efficient fine-tuning and deployment of visual foundation models, as demonstrated in the table below, which itself could be more insightful than simply achieving state-of-the-art performance.

| Method  | training-side resource efficiency | deployment-side inference efficiency |
| ------- | ------------------- | --------------------- |
| PEQA    | $\times$            | $\checkmark$          |
| QA-LoRA | $\times$            | $\checkmark$          |
| QST     | $\checkmark$        | $\times$              |
| Ours    | $\checkmark$        | $\checkmark$          |


### Reviewer wo2P (initial rating: 4)

After discussion, the reviewer has a remaining concern and suggests “**it would be valuable** to see comparative experiments regarding modeling channel correlations for further performance improvements.”
Following the reviewer’s suggestion, we have made a further exploration and analysis on channel correlations. The result suggests that designs that incorporate channel correlation have the potential for further improvement. We have added this further exploration in our revised paper (Appendix K), and we believe this remaining concern is now adequacy addressed.


### Reviewer XqgE (initial rating: 4)

The reviewer's main concerns stem from a **misunderstanding** of the difference between gradient checkpointing (GC) in our design and the regular GC operation. We have provided a more detailed clarification and the suggested analysis in our revised paper, which further highlight the role and effectiveness of our GC design. Additional suggestions on method explanation and efficiency analysis across different batch sizes have also been incorporated in our revised paper. Although this reviewer did not engage in the discussion, we believe the main concerns are now sufficiently addressed.


We thank the AC and all reviewers again for your efforts and valuable time, and we sincerely hope that our summary can provide clarity and assist with the final decision.

---

### Meta-Review · Area_Chair_X3MS · 2026-01-06

**Summary:**

This paper proposes a framework, termed EQuA, for joint quantization and adaptation for visual foundation models. Three reviewers out of five are relatively negative about this paper. The reviewers pointed out several critical concerns about different aspects of this paper, including but not limited to:

1. The motivation needs to be clarified. This paper lacks an overall description in the method section, particularly for the initial quantization and the training pipeline.
2. The cost of the weight importance calculation and SSA selection is missing. It would be better to provide the computational cost (e.g., time) and resource footprint (e.g., memory)
3. The practical advantage of using LoRA is questionable. As shown in Table 3, the significant reduction in trainable parameters does not result in savings in training time or memory, while slightly hurting performance.
4. The robustness of the SSA selection strategy requires further demonstration.
5. The discussion on gradient checkpointing is insufficient. It is recommended to include some discussions, such as the application to other reference methods,
6. The discussions on Q K layers and on the inference efficiency are unsatisfied.
7. The paper does not analyze the relation between sub-net design and error propagation. Therefore, it is hard to understand why the authors chose the proposed sub-net structure.
8. The task distribution might influence the weight split results.
9. The channel selection strategy lacks detailed ablation, such as performance–memory trade-off curves across channel split ratios p and comparisons of alternative importance proxies.
10. The robustness of alternating SAQF and BAQF is not well analyzed.

**Reviewer Concerns:**

The 2nd, 4th, 5th concerns are partially addressed, while the others are still outstanding.

**Reviewer Scores:**

None.

---

### Decision · Program_Chairs · 2026-01-26

Reject